# Herbarium specimen sequencing allows precise dating of *Xanthomonas citri* pv. *citri* diversification history

Paola E. Campos [1,2], Olivier Pruvost [1], Karine Boyer[1], Frederic Chiroleu[1], Thuy Trang Cao[1], Myriam Gaudeul[2,3], Cláudia Baider [4], Timothy M. A. Utteridge [5], Nathalie Becker[2,8], Adrien Rieux [1,8] & Lionel Gagnevin [6,7,8]

Herbarium collections are an important source of dated, identified and preserved DNA, whose use in comparative genomics and phylogeography can shed light on the emergence and evolutionary history of plant pathogens. Here, we reconstruct 13 historical genomes of the bacterial crop pathogen *Xanthomonas citri* pv. *citri* (*Xci*) from infected *Citrus* herbarium specimens. Following authentication based on ancient DNA damage patterns, we compare them with a large set of modern genomes to estimate their phylogenetic relationships, pathogenicity-associated gene content and several evolutionary parameters. Our results indicate that *Xci* originated in Southern Asia ~11,500 years ago (perhaps in relation to Neolithic climate change and the development of agriculture) and diversified during the beginning of the 13th century, after *Citrus* diversification and before spreading to the rest of the world (probably via human-driven expansion of citriculture through early East-West trade and colonization).

Plant pathogens have plagued human societies since the beginning of agriculture[1]. In such human-engineered ecosystems, the high density and low genetic diversity of hosts, as well as the environmental homogeneity caused by agricultural practices, facilitated the rise and propagation of diseases, with the evolution of host-adapted and virulent pathogens[2,3]. Intensification of agriculture, monoculture size rise, and trade globalization contributed to the emergence and expansion of pathogens, with opportunities to meet new naive host populations and realize host shift or host jump[3–5].

Today, plant pathogens and pests cause up to 40% yield loss in major crops, threatening food security[6], agrobiodiversity conservation, and public health[4,7]. A better understanding of the factors underlying the origin, evolution, and emergence of pathogens would help assess the risks they pose to crops and improve tools for surveillance and disease control. The combination of genetic material obtained from historic biological collections such as herbaria and modern samples provides heterochronous datasets which can improve phylogenetic estimates of evolutionary parameters and the timelines of their emergence and spread by bringing robust time components to inferences[8–10]. Indeed, adding ancient or historical sequences expands the temporal range of the dataset, increasing the chance to detect evolutionary change, i.e., temporal signal, which can be used to infer substitution rates and divergence time between lineages, as well as sudden modifications in genetic diversity[11–13].

[1]CIRAD, UMR PVBMT, F-97410 St Pierre, La Réunion, France. [2]Institut de Systématique, Évolution, Biodiversité (ISyEB), Muséum national d'Histoire naturelle, CNRS, Sorbonne Université, EPHE, Université des Antilles, 57 rue Cuvier, CP 50, 75005 Paris, France. [3]Herbier national, Muséum national d'Histoire naturelle, CP39, 57 rue Cuvier, 75005 Paris, France. [4]The Mauritius Herbarium, Agricultural Services, Ministry of Agro-Industry and Food Security, R.E. Vaughan Building (MSIRI Compound), Reduit 80835, Mauritius. [5]Royal Botanic Gardens, Kew, Richmond TW9 3AE, UK. [6]PHIM Plant Health Institute, Univ. Montpellier, CIRAD, INRAE, Institut Agro, IRD, Montpellier, France. [7]CIRAD, UMR PHIM, Montpellier, France. [8]These authors contributed equally: Nathalie Becker, Adrien Rieux, Lionel Gagnevin. ✉e-mail: adrien.rieux@cirad.fr; lionel.gagnevin@cirad.fr

The most well-studied crop pathosystem using historical herbarium genetic material is *Phytophthora infestans*, the oomycete responsible for potato late blight. Through the sequencing of 19th-century-infected specimens, the strain which caused the great potato famine in 1845–1849 has been identified and its genome characterized. Phylogeny reconstruction showed the historical strain to have originated from a secondary diversification area of the pathogen in North America from where one or a few dispersal events caused *P. infestans* emergence in Europe[14–19]. Similar studies reconstructing the evolutionary history of crop pathogens from full genomes have been successfully realized on viruses as well[20–23]. We recently described the history of the local emergence of the bacterial crop pathogen *Xanthomonas citri* pv. *citri* (*Xci*) in the South West Indian Ocean, using the first ancient bacterial genome retrieved from a herbarium specimen[24].

*Xci*, responsible for Asiatic citrus canker (ACC) and found in most subtropical citrus-producing regions, is a serious threat to citriculture. With no available definitive control measure, the disease causes important economic losses, by decreasing fruit yield and quality, and by resulting in strong restrictions on the commercial exchange of fruits and plants from infested regions[25,26]. *Xci* comprises three major pathotypes, discriminated by genetic diversity and host range. Pathotype A, with the broadest host-range (nearly all *Citrus* and several related rutaceous genera), is the most prevalent worldwide[27]. Pathotypes A* and A^W, primarily reported from Asia, are restricted to Key lime (*Citrus aurantiifolia*) and its close relative Alemow (*Citrus macrophylla*)[28,29]. Pathotype A* also occasionally infects Tahiti lime (*Citrus latifolia*) or sweet lime *(Citrus limettioides)*. A specificity of pathotype A^W is to elicit a hypersensitive response in several *Citrus* species, including *Citrus paradisi* and *Citrus sinensis*[30]. The phylogenetic relationships between the different pathotypes, first reconstructed using minisatellite molecular markers[31], before obtaining more resolutive data from whole genomes[32,33], suggested that pathotypes A and A^W are more closely related to each other than they are to pathotype A*. Recently, comparative genomic and phylogenomic analysis of 95 contemporary genomes were used to identify pathotype-specific virulence-associated genes and infer a probable scenario for *Xci* origin and diversification[34]. This study revealed that the origin of *Xci* occurred much more recently than the main phylogenetic splits of *Citrus* plants, suggesting dispersion, rather than host-directed vicariance, as the main driver of this pathogen geographic expansion. However, the sole use of modern genomes impeded the detection of sufficient de novo evolutionary change within the dataset (as referring to "measurably evolving populations"[12,35]). The authors were thus compelled to build a timeframe of evolution based on both the extrapolation of rates from external measures (i.e. rate dating) and a constraint on the distribution of a single external node age (i.e. node dating), two dating methodologies known to yield potential misleading estimates[8,36].

In the present study, we took advantage of an extensive collection of contemporary *Xci* strains sampled in the field during the last 70 years along with a broad representation of *Citrus* specimens in herbaria dating back to the 19th century. We sequenced historical bacterial genomes of *Xci* from 13 herbarium samples showing typical canker symptoms and originating from the putative center of origin of the pathogen. Authentication based on DNA degradation patterns was established, allowing us to identify, in more detail, a significant contribution of sample age and library production protocol to the deamination rate. We then compared the historical genomes to those of 171 modern strains representative of worldwide genetic diversity, 57 of which were specifically sequenced for the purpose of this study. We aimed to improve knowledge about *Xci* origin and diversification history by (1) reconstructing a thorough time-calibrated phylogeny and inferring evolutionary parameters based on a robust dating approach within the measurably evolving populations framework, (2) inferring the ancestral geographical state of lineages and estimating source

populations of epidemics, (3) assessing the pathogenicity-associated gene content across all lineages.

## Results

### Laboratory procedure and high-throughput sequencing
Thirteen herbarium samples were processed into libraries, using a TruSeq Nano (Illumina) protocol for seven of them, and a homemade BEST protocol for the six others[37]. Sequencing produced between 56.3M and 365.2M paired-end reads. Following quality checking and adapter trimming, reads were merged (accounting for 96.8–99.8% of controls passing reads), presenting insert median lengths of 32–92 nt (mean lengths of $42.1 \pm 12.8$ to $102.9 \pm 45.1$ nt).

### Historical genomes reconstruction and ancient DNA damage pattern assessment
Thirteen historical drafts of *Xci* genomes were reconstructed by mapping such processed and merged reads on reference strain IAPAR 306 genome (chromosome, plasmids pXAC33 and pXAC64)[38], with one of them, HERB_1937, processed previously[24]. The proportion of reads mapping to the *Xci* reference genome ranged from 0.82% (HERB_1937) to 27.10% (HERB_1922) (Table 1). Importantly, no *Citrus* nor *Xci*-specific DNA fragments were found in our negative *Coffea* sp. controls, thus ruling out in-lab contamination.

Coverage (proportion of reference genome covered) at $1X$ ranged between 94.6% and 98.2% for the chromosome, and 49.7–97.1% for the plasmids (Table 1 and S2), due to the fact that plasmid contents are usually highly variable[39]. By contrast mean depths (average number of mapped reads at each base of the reference genome) were between $6.2X$ and $96.2X$ for the chromosome and between $17.9X$ and $130.3X$ for the plasmids, reflecting the presence of multiple copies of plasmids in *Xanthomonas* (Table 1 and S2).

Ancient DNA typically presents short fragments and cytosine deamination at fragment extremities[40]. We analyzed such degradation patterns using the dedicated tool mapDamage2[41]. For reads aligning to the chromosome sequence, herbarium samples displayed mean fragments lengths of $41.1 \pm 9.3$ to $88.1 \pm 39.2$ nt, and 3'G > A substitution rates at terminal nucleotides of 1.22–3.70%, decreasing exponentially along the DNA molecule for all historical genomes. Examining 5'C > T substitution rates gave similar results (Table 1 & Fig. 1).

Similar patterns were obtained for plasmid-like sequences (Fig. S1). Collection time (i.e. storage length) has been suggested to influence deamination rates, as shown for more than 50 herbarium samples over a 1750–2000 time span[42], so we examined this possibility. Figure S2 shows the first "protocol effect" on deamination rates ($p < 0.0001$), which had to be taken into account. Thus, we performed our analysis within each protocol subgroup, covering similar timespans (7 samples collected from 1845 to 1974, versus 6 samples collected from 1852 to 1963), and observed a significant, time-dependent, effect on terminal deamination rates ($p < 0.0001$). Experimental validation of the protocol effect was neither the aim of this study, nor possible, due to the scarcity of symptom-harboring plant material.

Finally, within each library construction protocol, 3'G > A deamination rates were significantly higher for plasmid-type sequences (Fig. S2, $p < 0.0001$ for BEST protocol and $p = 0.0259$ for TruSeq Nano protocol). Examining 5'C > T substitution rates gave the same results. In the first published HERB_1937 herbarium sample, a significantly higher terminal substitution rate was observed for plasmid versus chromosome sequences, irrespective of fragment length[24]. In this study, in addition to deamination rates per se, the total number of analyzed reads, as well as 11 fragment length classes, were taken into account for each sample, within each library protocol (BEST, TruSeq Nano) and within each sequence type (plasmid, chromosome) (Fig. S1 and Table S1). All the 26 (but one) plasmid-type sequences harbored higher deamination rates, as compared to their respective chromosome-type sequences.

**Table 1 | Summary of mapping, depth, coverage, and damage statistics for the 13 historic *Xci* genomes**

| ID | Protocol | Million reads | Merged reads (%) | *Xci* reads (%) | Chromosome | | | |
|---|---|---|---|---|---|---|---|---|
| | | | | | Depth (*X*) | Coverage at 1*X* (%) | Insert length (mean ± SD, nt) | Deamination rate at terminal position (%) |
| HERB_1845 | TruSeq Nano | 414.3 | 97.78 | 10.39 | 82.1 | 98.3 | 50.4 ± 23.0 | 3.68 |
| HERB_1884 | TruSeq Nano | 246.8 | 97.82 | 10.48 | 55.6 | 98.1 | 46.2 ± 16.6 | 3.22 |
| HERB_1911 | TruSeq Nano | 365.2 | 99.50 | 2.05 | 32.4 | 98.2 | 69.1 ± 22.3 | 3.70 |
| HERB_1915 | TruSeq Nano | 217.3 | 97.12 | 6.04 | 39.3 | 98.1 | 47.9 ± 17.9 | 2.81 |
| HERB_1937 | TruSeq Nano | 220.9 | 98.89 | 0.82 | 6.2 | 94.6 | 42.7 ± 12.7 | 3.65 |
| HERB_1946 | TruSeq Nano | 262.5 | 98.00 | 7.52 | 64.3 | 98.2 | 57.9 ± 29.5 | 1.81 |
| HERB_1974 | TruSeq Nano | 260.8 | 96.97 | 6.80 | 35.9 | 97.9 | 41.1 ± 9.3 | 2.09 |
| HERB_1852 | BEST | 314.9 | 99.97 | 4.81 | 63.7 | 97.8 | 70.9 ± 43.6 | 1.89 |
| HERB_1854 | BEST | 113.0 | 99.96 | 10.76 | 54.1 | 98.3 | 75.7 ± 42.8 | 1.22 |
| HERB_1859 | BEST | 159.5 | 99.83 | 5.31 | 41.8 | 97.7 | 73.9 ± 33.8 | 1.76 |
| HERB_1865 | BEST | 156.5 | 99.04 | 4.91 | 49.8 | 98.2 | 88.1 ± 39.2 | 1.57 |
| HERB_1922 | BEST | 120.9 | 99.66 | 27.10 | 96.2 | 97.3 | 67.9 ± 29.3 | 1.22 |
| HERB_1963 | BEST | 56.3 | 99.89 | 14.63 | 43 | 98.0 | 74.7 ± 32.3 | 1.25 |

*SD* standard deviation, *nt* nucleotides.

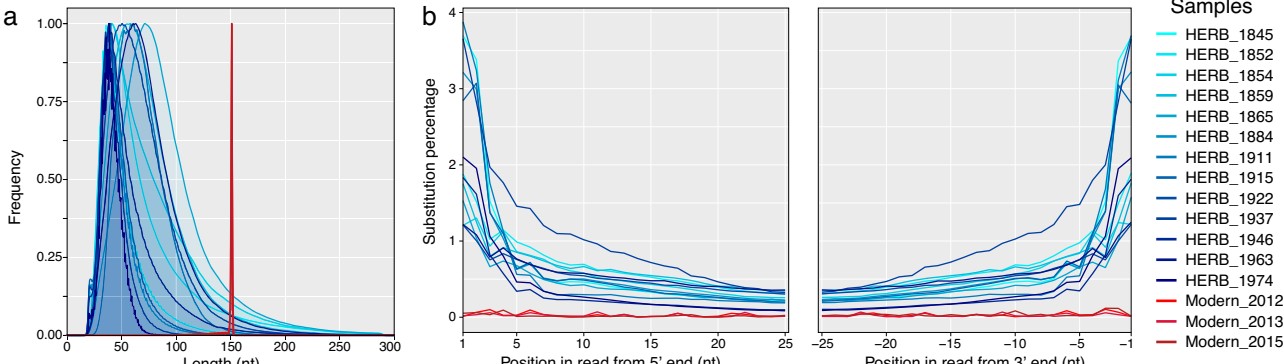

**Fig. 1 | *Post-mortem* DNA damage patterns measured on reads mapping to the *Xci* chromosome. a** Fragment length distribution (nt: nucleotides; relative frequency in arbitrary units). **b** Substitution percentage of the first 25 nucleotides of R2 reads (5′ C to T substitutions, left panel), complementary to the last 25 nucleotides of R1 reads (3′ G to A substitutions, right panel) of the 13 historical genomes (blue lines, light to dark gradient from the oldest to the youngest) and three modern *Xci* strains (red lines, light to dark gradient from the oldest to the youngest). Source data are provided as a Source Data file.

## Modern genomes reconstruction

A total of 171 modern genomes (17 modern strains from pathotype A*, 4 from pathotype A^W and 150 from pathotype A) were also reconstructed by mapping reads on the reference strain IAPAR 306 genome. The proportion of reads mapping to *Xci* reference genome averaged 93.5% and ranged from 61.6% to 99.9%, as expected for DNA extracted from freshly cultured bacteria. Coverage at 1*X* ranged between 96.1% and 100.0% for the chromosome, and 54.3–100.0% for the plasmids. Mean depths ranged between 35.8*X* and 244.8*X* for the chromosome and between 40.6*X* and 247.2*X* for the plasmids. Mean fragments length mirrored the number of bases sequenced (150 bp) and modern DNA controls displayed no deamination patterns (Fig. 1).

## Phylogenetic reconstruction, dating and ancestral geographic state estimation

Alignment of the chromosome sequence of the 13 historical genomes and 171 modern genomes (17 modern strains from pathotype A*, 4 from pathotype A^W and 150 from pathotype A) allowed for the identification of 15,292 high-quality single nucleotide polymorphisms (SNPs). ClonalFrameML identified four major recombining regions (Table S2), from which 2285 SNPs were removed from further inferences. On the 13,007 recombination-free SNP alignment, a

paraphyletic outgroup was added, formed of *X. a.* pv. *vasculorum* NCPPB-796 and two strains phylogenetically close to *Xci*, *X. c.* pv. *cajani* LMG558 and *X. c.* pv. *clitoriae* LMG9045. A maximum-likelihood (ML) phylogeny was built with RAxML[43] and rooted with *X. a.* pv. *vasculorum* (Fig. S3). Strains from each pathotype grouped together and formed distinct clades. Clade A* was at the root of clade *Xci*, while clade A^W was a sister group of clade A. This topology (A*, (A^W, A)) was highly supported with bootstraps values of 100. Clade A displayed three major, highly supported lineages which we named A1–A3: lineage A1 corresponds to the main group of strains (lineage A in Patané et al.[34], DAPC1 in Pruvost et al.[31]), contained 10 historical specimens, and has a polytomic structure. Lineage A2 contains strains from India and Pakistan but also from Senegal and Mali and corresponds to lineage A2 as described by Patané et al.[34]. Lineage A3 contains seven strains from Bangladesh as well as three historical specimens from Bangladesh, India, and China (Yunnan). Lineages A2 and A3 both contain strains classified as DAPC2 by Pruvost et al.[31]. Within these lineages, strains are mostly grouped according to their geographic origin, with a few exceptions and with Asia being represented in all groups and subgroups (Fig. S6). Historical specimens were mainly clustered with modern strains of the same geographical origin.

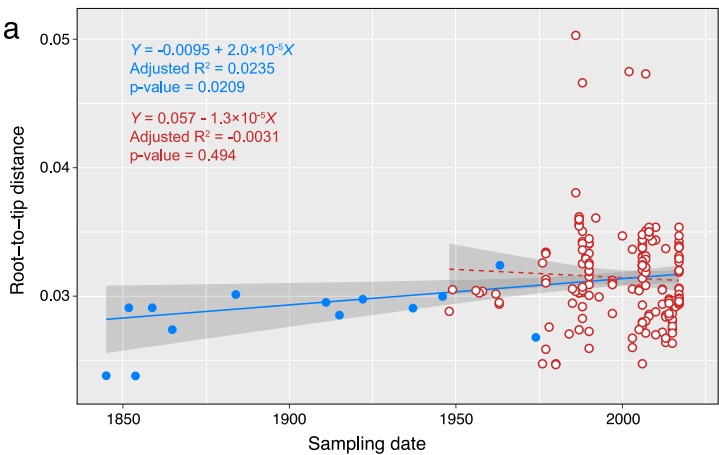

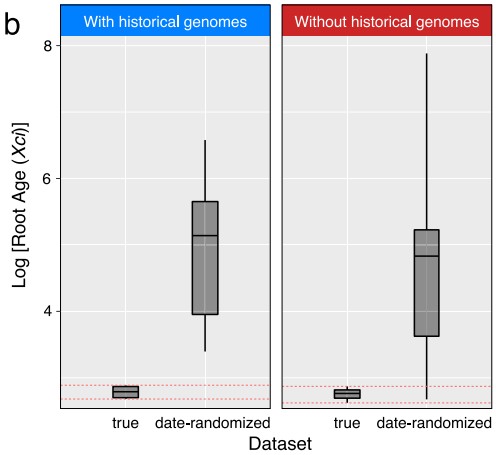

**Fig. 2 | Root-to-tip regression and date-randomization temporal test. a** Root-to-tip linear regression lines plotted either in blue solid line when integrating historical genomes (blue dots, $n = 184$), and in red dashed line when only computed from modern genomes (white dots with red line, $n = 171$). Gray areas (error bands) indicate 95% confidence intervals. Associated values are the linear regression equation, adjusted $R^2$ and $p$-value obtained from a two-sided Student test under the null hypothesis of a slope equal to zero, with ($n-1$) degrees of freedom

(significant only when including historical genomes). **b** Evaluating temporal signal in the dataset by date-randomization test showed no overlap between the age of the root estimated from the real and date-randomized datasets (summed up in a single box) when historical genomes were included (left). Boxplots display the five following summary statistics: minimum, first quartile, median, third quartile and maximum of the 95% highest posterior density intervals. Statistics were computed from $n = 10,000$ iterations. Source data are provided as a Source Data file.

The ML tree was used to test the presence of temporal signal (i.e., progressive accumulation of mutations over time) within the *Xci* clade using three different tests. The linear regression test between root-to-tip distances and sampling ages displayed a significantly positive slope (value = 0.00002, adjusted $R^2 = 0.0235$ with a $p$-value = 0.0209) (Fig. 2a). Interestingly, such a pattern was also conserved at several other internal nodes (Fig. S4). Second, the date-randomization test was successfully passed as the inferred root age from the real *versus* date-randomized datasets exhibited no overlap of the 95% highest posterior density (HPD) (Fig. 2b). Finally, the Mantel test displayed no confounding effect ($r = -0.73$, $p$-value = 0.89) between temporal and genetic structures. To specifically evaluate the contribution of historical genomes to the magnitude of temporal signal, we repeated the above tests on a dataset containing modern genomes only, producing no temporal signal at the *Xci* clade scale (Fig. 2a, b).

A Bayesian time-calibrated tree was built with BEAST (Fig. 3), and found globally congruent (similar topology and node supports) with the ML tree. The root of the *Xci* clade (node at which *Xci* diversified into numerous pathotypes) was inferred to date to at least 1437 CE (current era), possibly being even older, at most 962 CE (95% HPD), with an average around 1218 CE.

We obtained a mean substitution rate of $14.30 \times 10^{-8}$ [95% HPD: $12.47 \times 10^{-8} - 16.14 \times 10^{-8}$] per site per year with a standard deviation for the uncorrelated log-normal clock of 0.507 [95% HPD: 0.428−0.594], suggesting low rate heterogeneity among tree branches. The inferred dates of other internal nodes of interest, including the MRCA for each of the three pathotypes, as well as for some geographically structured clades, are given in Table 2.

To date the origin of *Xci*, i.e., the date at which it diverged from its closest relatives, a secondary alignment was realized with the 13 historical sequences, the 171 modern sequences and the three outgroup sequences. A total of 209,306 chromosomal high-quality SNPs was found, 198,249 of which were outside recombining regions. The presence of temporal signal was tested as previously described. On this dataset, temporal signal was still present at the root of *Xci* clade but the signal disappeared at the external node connecting the outgroup, with *X. c.* pv. *cajani* as *Xci* closest relative (root-to-tip regression test: slope value = $-4.6 \times 10^{-6}$, adjusted $R^2 = 0.0006$ with a $p$-value = 0.293; date-randomization test: overlapping of the 95% HPD of the root age (Fig. S5); Mantel's test: $r = 0.04$, $p$-value = 0.09). As the prerequisite for

tip-dating was not met, we performed a rate-dating analysis integrating the mean substitution rate inferred in the tip-dating calibration. The node date at which *Xci* split from other *Xanthomonas citri* pathovars was inferred to −9501 CE [95% HPD: −13099 to −6446].

In order to infer ancestral location state at nodes, geographic signal in the dataset must first be detected. By measuring the association index between topology and the location trait data on both the real tree and 1000 location-randomized trees, our results highlighted the existence of a non-random association between spatial and phylogenetic structure ($p$-value < 0.0001). A discrete phylogeographic analysis was therefore run with BEAST (Fig. 3, Table 2), inferring a highly supported South Asia 1 (Bangladesh, India and Nepal) origin where *Xci* split from its *Xanthomonas citri* relatives and then diversified through its three lineages A*, A^W^ and A. All of them, comprising A1−A3 lineages included in A, were also inferred to have diversified in the same South Asia 1 region. In the lineage A1, the polytomy was inferred to have a Southeast Asia 2 (Indonesia or Philippines) origin (node support of 1, state posterior probability of 0.62). Southeast Asia 2 origin was also inferred for the lineages composed of: (1) all New Zealand strains (herbarium specimen HERB_1946_A_Guam at the root); (2) all Argentina strains (with HERB_1854_A_Indonesia and HERB_1845_A_Indonesia branching closely), and (3) nearly all strains from the SWIO islands and from Martinique (with HERB_1974_A_Mauritius and HERB_1937_A_Mauritius at their root).

## Pathogenicity-associated genes content

We investigated the presence or absence of 144 pathogenicity-associated genes (Supplementary Data 2) under a mapping approach. Only 10 identified virulence factors were variable between strains of *Xci*: *xopAF*, *xopAG*, *xopAV*, *xopC1*, *xopE*, *xopT*, XAC1496, XAC2265/*helD*, XAC3278 and XAC3294 (Fig. 3 and Supplementary Data 3). The twenty-four genes coding for the type III secretion system (T3SS) were present in all *Xci* strains. Of the 66 type III effectors (T3E) genes tested, we found 32 present and 28 absent in all *Xci* strains, whereas 6 were of variable presence (*xopAF*, *xopAG*, *xopAV*, *xopC1*, *xopE*, *xopT*), in accordance with previous works. Variation in some virulence factors (for example, *xopE2*, *xopAV*, XAC3294) was anecdotal, concerning only one or a few strains; in other cases, their distribution was found to be more or less clade-dependent: *xopAG* is

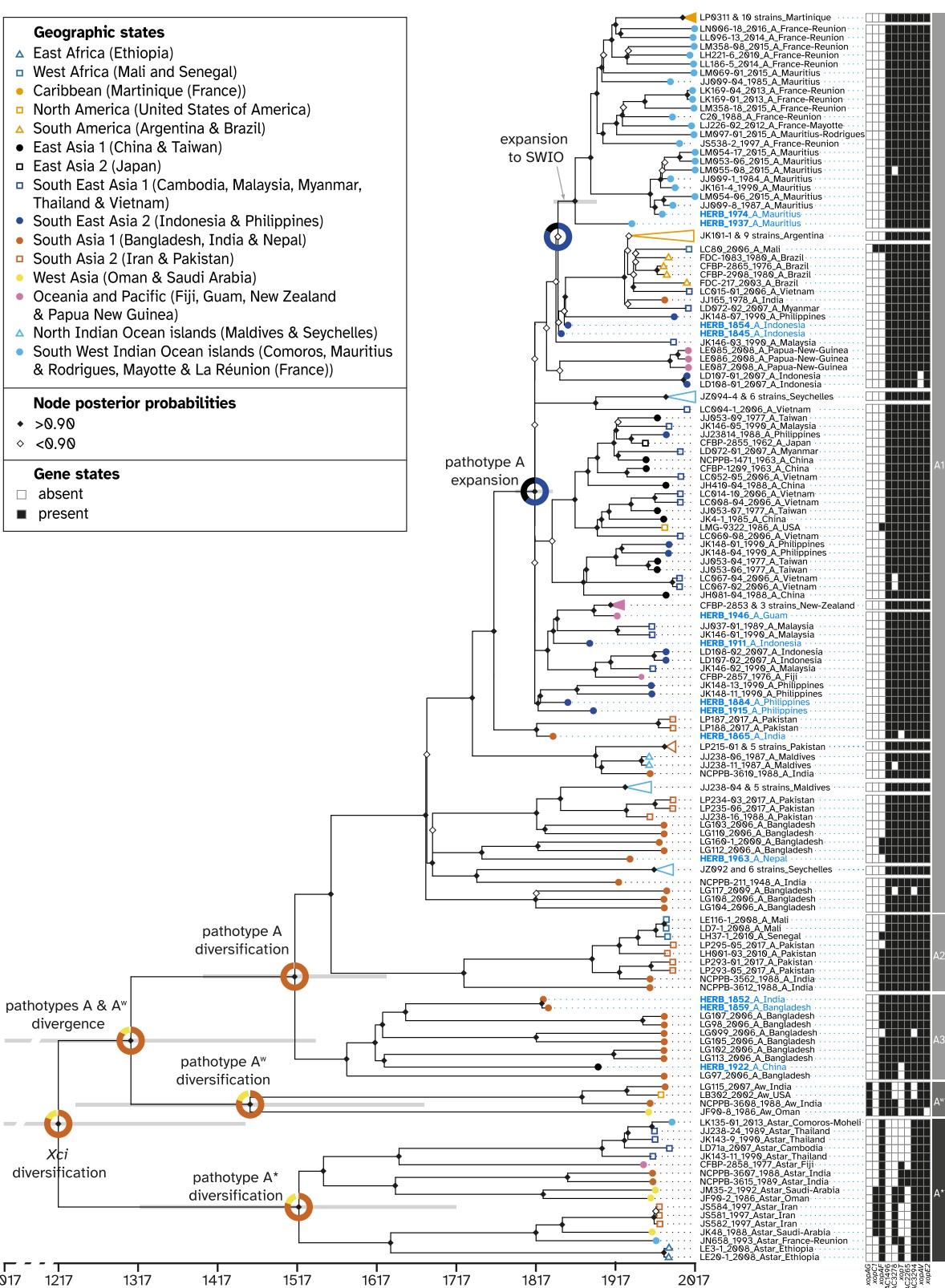

**Fig. 3 | Spatiotemporal Bayesian reconstruction of *Xci* evolutionary history.**
Dated phylogenetic tree including 13 historical specimens (blue labels) and 171 modern strains (black labels) built from 13,007 recombination-free SNPs. Node support values are displayed by diamonds; node bars cover 95% highest probability density of node height. Branch tips are colored according to the sample's geographic origin. Groups of closely related strains were collapsed for better visibility; details of each group can be found in Fig. S2. Reconstructed ancestral geographic states are represented at some nodes of interest with pie charts representing the posterior probability of geographical regions as the origin of the node. Strain labels include strain name, collection year, pathotype, and country of origin. Presence and absence of 10 variably present pathogeny-associated genes are indicated next to strain labels by black squares and white squares, respectively. Pathotypes and lineages are indicated to the right. Source data are provided as a Source Data file.

**Table 2 | Inferred spatiotemporal data at major nodes**

| Node | Date (CE) [95% highest probability density] | Location (posterior probability) |
|---|---|---|
| *Xci* origin | −9501 [−13099 to −6446] *[−12052 to −3851]* | South Asia 1 (0.85) *Bangladesh, India, Pakistan* |
| *Xci* diversification | 1218 [962–1437] *[−3648 to 285]* | South Asia 1 (0.81) *Bangladesh, India, Pakistan* |
| *Xci* (A^W, A) divergence | 1305 [1061–1511] | South Asia 1 (0.82) |
| *Xci* A* diversification | 1523 [1377–1650] | South Asia 1 (0.80) *Iran, Oman, Saudi Arabia* |
| *Xci* A^W diversification | 1461 [1227–1675] | South Asia 1 (0.80) *Bangladesh, India, Pakistan* |
| *Xci* A diversification | 1524 [1390–1637] | South Asia 1 (0.99) *Bangladesh, India, Pakistan* |
| *Xci* A - (A2, A1) divergence | 1570 [1408–1684] | South Asia 1 (1.00) *Bangladesh, India, Pakistan* |
| *Xci* A1 diversification | 1696 [1607–1768] | South Asia 1 (1.00) *Bangladesh, India, Pakistan* |
| *Xci* A expansion (polytomy) | 1829 [1799–1845] | South East Asia 2 (0.62) |
| *Xci* A—New Zealand MRCA[a] | 1925 [1896–1949] | South East Asia 2 (0.79) |
| *Xci* A—South America MRCA[a] | 1930 [1907–1951] | South East Asia 2 (0.96) |
| *Xci* A—SWIO islands MRCA[a] | 1868 [1844–1899] | South East Asia 2 (0.83) |
| *Xci* A—Caribbean MRCA[a] | 2000 [1991–2008] | SWIO islands (1.00) |

Estimations inferred by Patané et al. [34] (in italics) were indicated when possible.

*CE* Current Era, *SWIO* South West Indian Ocean, South Asia 1: Bangladesh, India and Nepal; South East Asia 2: Indonesia and Philippines; SWIO islands: Comoros, Mauritius, Rodrigues, Mayotte and La Réunion.

[a]For nodes indicated with an asterisk, reconstructed locations are given for the parent node, i.e., the split between the MRCA of the strains from the indicated geographic areas and its closest relative.

present only in pathotype A^W strains, as expected[44]. *XopC1* is present in a few A* strains[34]. XAC1496 is specifically absent from all A* strains[34], suggesting an acquisition during the pathotype differentiation process. *XopAF* is present only in the deeply rooted branches of the tree, suggesting a loss of the effector for strains of clade A1. XAC2265 is absent from A* strains except for two of them, suggesting reacquisition of this factor. Conversely, XAC3278 and *xopT* seem to have been lost several times in the recent past. *XopF1* is pseudogenized, and probably not functional[45]. Genes encoding for transcription activator-like effectors (TALE) are present in all strains (homologs of *avrBs3* in Supplementary Data 2), but because of the chosen sequencing strategy in our study, which produces short reads, the paralog number in each strain and the complete sequence of each paralog could not be reconstituted and compared due to their repetitive nature (multiple paralogs and central region of the gene composed of more than ten 102-bp repeats). Nevertheless, at least one functional homolog of the TALE PthA4 must be present in each strain, as modern strains and herbarium nucleic acids were isolated from canker symptoms, which is caused by PthA4[46,47]. In addition, among 54 genes potentially involved in virulence but not necessarily dependent on the T3SS, only 4 were of variable presence in *Xci* (XAC1496, XAC2265/*helD*, XAC3278 and XAC3294), the 50 others being systematically present. In general variation was localized in the basal branches of the tree, and concerned strains from pathotypes A* and A^W. Conversely, strains belonging to pathotype A were rather homogenous in their pathogenicity factor contents.

Inconsistencies with other works can be due to our identification strategy (mapping on more than 75% of the CDS). For example, XAC3278 (XAC_RS16605) was considered to be present in all strains by Patané et al. [34], but we found that there can be two paralogs of the gene. One is distant (65% nucleotidic identity) from the functionally demonstrated copy but is present in all strains, while the other, 100% homologous to XAC3278, is only present in some strains. Similarly, we found variability in the length of homologs to XAC2265 (XAC_RS11510) between strains, some being truncated of more than 30% of their length, resulting in variable presence.

## Discussion

In this study, we successfully reconstructed the genome of 13 *Xci* historical strains from herbarium material collected between 1845 and 1974, which we compared with a set of 171 modern genomes representative of the bacterial global diversity, 57 of them having been specifically generated for this study. A better understanding of *Xci* evolutionary history is a subject of great interest since it may help deciphering how bacterial pathogens specialize on their hosts and diversify while expanding their geographical range.

At a molecular level, the analysis of historical genomes was highly informative. First of all, assessment of *post-mortem* DNA degradation patterns specific to ancient DNA, such as fragmentation and deamination, confirmed the historical nature of the reconstructed genomes. Terminal deamination values were consistent with those of Weiss et al. [42] obtained from herbarium samples (roughly, between 1.5% and 5.0%). Deamination rates were found higher for plasmidic versus chromosomal sequences, possibly due to chromosome-specific cytosine methylation patterns (discussed in Campos et al. [24]). We finally showed an age-dependent deamination for bacterial DNA from herbarium material, as described elsewhere for nuclear and chloroplastic plant DNA[42].

Adopting a shotgun-based deep sequencing strategy revealed between 0.8% to 27.1% of *Xci* DNA amongst the 13 historical samples, a wide variation falling within the range of previous studies that attempted to retrieve non vascular pathogen DNA from infected herbarium leaves[14,18,19]. We aligned the 13 historical genomes with 171 modern representatives of the bacterial global diversity, and built a phylogenetic tree from the chromosome-wide non-recombining SNPs. This tree confidently associated the pathotypes to be monophyletic groups, and displayed a (A*, (A^W, A)) topology, as previously reported from genome-wide SNPs[33] and unicopy gene families analyses[34]. Interestingly, the relationships inside the pathotype A (A3, (A2, A1)) agreed with the former analysis but not the latter, whose discrepancy could be explained by the under-representation of A3 lineage (only LG98_2006_A_Bangladesh present). Globally, the observed geographic clustering inside the pathotype A clade is

consistent with previous studies[31,34,39]. Clade A1 had a somewhat polytomic structure (most lineages seem to have branched from a single ancestor at the same moment). This can either be the result of a rapid and synchronous expansion of *Xci* in contrasted environments, leading to the divergence and maintenance of multiple lineages, or of an artifact of building topologies with insufficient data, when lack of information does not allow to differentiate distinct divergence events[48]. Our dating (see below) of the beginning of the A1 diversification fits with the first scenario, but as many nodes in the A1 cluster are not well supported (white diamonds in Fig. 3), we cannot fully exclude the second one.

The presence of temporal structure is an essential prerequisite to perform tip-calibrated inferences[11–13]. While the dataset containing contemporary genomes only (1948−2017) did not reveal the existence of any measurably evolving population, inclusion of the 13 historical genomes (1845−1974) brought the required temporal signal within the *Xci* clade. This allowed us to build a time-calibrated phylogeny without making any underlying assumption on the age of any node in the tree, nor on the rate of evolution in order to propose new evolutionary scenarios for the origin and diversification of the pathogen. Previous studies did not use historical strains[34], focused on a recent and local emergence[24], or were limited by the exploitation of a few partial genetic markers only[49].

We inferred a mean substitution rate of $14.30 \times 10^{-8}$ [95% HPD: $12.47 \times 10^{-8} - 16.14 \times 10^{-8}$] substitutions per site per year, a value ~1.5× faster than the one ($9.4 \times 10^{-8}$ [95% HPD: $7.3 \times 10^{-8} - 11.4 \times 10^{-8}$]) previously obtained on a single lineage within the pathotype A clade of *Xci*, at the local scale of the South West Indian Ocean islands[24]. We dated the MRCA of all *Xci* strains, the node leading to bacterial diversification, to the beginning of the 13th century (1218 CE [95% HPD: 962−1437]), a much more recent timespan than the one [−3648 to 285] inferred previously[34]. The discrepancy between those estimates probably arises from differences in the considered molecular dating methodologies. Indeed, their molecular clock was calibrated by applying a prior on both the rate of evolution (estimated on housekeeping genes of non-*Xci Xanthomonas* species) and the age of a node external to the *Xci* clade (indirectly deriving from the same rate of evolution), while the methodology used in our work makes use of the age of the strains only, a method shown to yield far more accurate and robust estimates[8,13,36]. In addition of dating the emergence of the three *Xci* lineages A*, A^W & A, we also inferred the ones of geographically structured lineages such as the one in New Zealand in 1925 [95% HPD: 1896−1949], in South America in 1930 [95% HPD: 1907−1951] or in Martinique in 2000 [95% HPD: 1991−2008] with values always predating disease first reports made in 1937[50], 1957[51], and 2014[52], respectively. Finally, as the divergence between a pathogen and its closest known relative places a maximum bound on the timing of its emergence[10], we included three outgroup sequences, of which *X. c.* pv. *cajani*, a pathogen of the *Fabaceae* plant family, was the first to branch out of the *Xci* clade. As the inclusion of divergent outgroup genomes precluded the application of tip-dating methodology we extrapolated the rate of substitution previously estimated within the *Xci* clade to date the split between *Xci* and *X. c.* pv. *cajani* to −9501 CE [95% HPD: −13099 to −6446], a value which partly overlap with the inferred interval of [−12052 to −3851] found by Patané et al.[34], although on a more constrained length of time.

Our phylogeographic analysis inferred an origin and diversification of *Xci* in an area of South Asia neighboring to Bangladesh, India and Nepal, consistently with previous reconstructions and estimations based on genetic diversity[31,33,34]. It also corresponds to the area of origin of the *Citrus* genus, which is believed to have emerged within the southern foothills of the Himalayas (including Assam, Western Yunnan and Northern Myanmar) 6−8 million years ago[53]. Our temporal calibrations indicate a mean age of 11.5 ky for the origin of *Xci*, a period which coincides with the beginning of the Holocene (−9700 CE

to present)[54] following the Bølling-Allerød warming global event (-−12700 CE to −10,900 CE)[55]. Such warmer and wetter climates could have facilitated plant expansion into new areas previously occupied by ice such as the mountainous regions and the northern parts of South Asia[56]. This was followed by the development of societies and agriculture in Northern India and in China during which movements of plants between and outside these regions may have gathered favorable conditions for *Xci* emergence on *Citrus* through bacterial host jump, as previously proposed[34]. The diversification of *Xci* was dated to the early 13th century in its area of origin, which was crossed at the time by the Southern Silk Road[26], linking Eastern and Western civilizations through trading. The westward commerce of goods, including citrus, which have been found in Mediterranean countries since −500 CE[57], as well as the breeding of citrus varieties for cooking and for raw eating[26], could have dispersed and isolated the pathogen into its three known pathotypes. More recent global changes observed between the 17th and 20th centuries, such as the spice trade and the development of a worldwide colonial agriculture might also be important factors in the apparently intense diversification of genotypes (and their global spread)[24].

We based our analysis of the contents in virulence-related genes on lists of proven and hypothetical factors, a large part of which are T3SS elements and T3E. As expected, and as reported previously all *Xci* strains contained genes for a full type III secretion apparatus and for a large number of T3E: with 32 genes it is larger than previously reported[58], mostly because new (often hypothetical) genes were described since then[59]. Variation was detected for six T3E among *Xci*. Factors not related to the T3SS were also assessed: we confirmed that most are present in all of the strains, with variation concerning four of them.

Herbarium samples were not different in their effector contents from their close phylogenetic relatives, indicating that most of the effector shuffling process occurred before the 1850s. This is consistent with evolutionary hypotheses that postulate that fundamental factors of pathogenicity (in our case the type III secretion apparatus and probably many non-T3 factors) are acquired early in the evolutionary history of plant-associated bacteria, providing a general adaptation to interactions with plants, with a subsequent host specialization (and differentiation in further clades) correlated to variable T3E contents and coevolutionary arms race[60−62]. Effectors that are usually the most variable within a *Xanthomonas* pathovar are TALE[63] but their repeated nature and the fact that DNA fragments from herbarium samples are fragmented do not permit their reconstruction.

Our work presents two main limitations. First, although with 163 genomes our dataset displayed the best representation of pathotype A genetic diversity published to date, the reconstructed phylogenetic tree exhibits high level of imbalance (with only 17 A* and 4 A^W genomes), a property previously shown to lead to reduced accuracy or precision of phylogenetic timescale estimates[64]. Bias in representation of populations, such as overrepresentation of one compared to the others or the absence of representatives from the true founder lineage, can also lead to the reconstruction of ancestral state tending to correspond to the oversampled population rather than the true founder lineage[65]. Although this feature arises from the fact that *Xci* worldwide expansion mostly involved pathotype A strains[31], future work should aim to better characterize the genomic diversity of A* and A^W strains. Secondly, as gene content variation analysis was performed by mapping reads to reference sequences, we were unable to identify potential genomic rearrangements among strains, a process known to be frequent within *Xanthomonas* species[61,66,67]. Similarly, this approach impeded us from identifying genetic content absent from the reference sequences. To overcome those limitations and better recover pathotype-specific genes, comparative genomic analysis based on de novo assembly and/or without a priori on the targeted genes would be interesting to perform.

**Table 3 | General characteristics of the 13 herbarium specimens**

| ID | Herbarium code | Herbarium specimen ID | Collection year | Location | Host |
|---|---|---|---|---|---|
| HERB_1845 | P | P05297986 | 1845 | Indonesia, Java | *Citrus aurantiifolia* |
| HERB_1852 | K | Q1874 | 1852 | India, Khasi hills | *Citrus medica* |
| HERB_1854 | K | Q1954 | 1854 | Indonesia, Java | *Citrus aurantiifolia* |
| HERB_1859 | P | P05240716 | 1859 | Bangladesh | *Citrus medica* |
| HERB_1865 | K | Q1889 | 1865 | India | *Citrus medica* |
| HERB_1884 | K | 1206 | 1884 | Philippines, Luzon | *Citrus medica* |
| HERB_1911 | P | P05297996 | 1911 | Indonesia, Java | *Citrus aurantiifolia* |
| HERB_1915 | P | P05297992 | 1915 | Philippines | *Citrus* sp. |
| HERB_1922 | US | 1756364 | 1922 | China, Yunnan | *Citrus medica* |
| HERB_1937 | MAU | MAU0015151 | 1937 | Mauritius | *Citrus* sp. |
| HERB_1946 | BPI | 686249 | 1946 | Guam, Tlofofo | *Citrus* sp. |
| HERB_1963 | K | 630116 | 1963 | Nepal, Sanichare | *Citrus medica* |
| HERB_1974 | MAU | MAU0015154 | 1974 | Mauritius | *Citrus* sp. |

To conclude, our study emphasizes how historical genomes from herbarium samples can provide a wealth of genetic and temporal information on bacterial crop pathogens evolution. Similar studies could be applied to other plant pathogens to infer the temporal dynamic of their populations and elucidate their evolutionary history with more resolutive estimations, which in turn may provide clues to improve disease monitoring and achieve sustainable control.

## Methods

### Herbarium material sampling

The collections of the Royal Botanic Gardens, Kew (K) (https://www.kew.org/science/collections-and-resources/collections/herbarium), the Mauritius Herbarium (MAU) (https://agriculture.govmu.org/Pages/Departments/Departments/The-Mauritius-Herbarium.aspx), the Muséum national d'Histoire naturelle (P) (https://www.mnhn.fr/fr/collections/ensembles-collections/botanique), the US National Fungus Collections (BPI) (https://nt.ars-grin.gov/fungaldatabases/specimens/specimens.cfm) and the U.S. National Herbarium (US) (https://collections.nmnh.si.edu/search/botany/) were prospected between May 2016 and October 2017. For each *Citrus* specimen displaying typical Asiatic citrus canker lesions, sampling permission was requested from the collection curator. When accepted, sampling was performed on site using sterile equipment and transported back to the laboratory in individual envelopes where they were stored at 17 °C in vacuum-sealed boxes until use. Thirteen historic specimens collected between 1845 to 1974 and conserved in five different herbaria were selected for analysis. Those were chosen as the oldest available from Asia, the supposed geographic origin of *Xci*, as well as from Oceania and the South west Indian Ocean. Among the 13 herbarium specimens, 12 were processed during the course of this study, while HERB_1937 has been processed previously[24]. A dataset listing the historical samples analyzed along with their metadata (herbarium code, herbarium specimen ID, host species, collection year, and location) was built using Microsoft Excel v16.71 (Table 3).

### Ancient DNA extraction and library preparation

DNA extraction from herbarium samples was performed in a bleach-cleaned facility room with no exposure to modern *Xci* DNA, following a custom CTAB protocol, along with blank extractions (sample buffers only) and herbarium samples of *Coffea* sp., a non *Xci*-host species included as negative control. Pools of five canker lesions (to obtain ~10 mg) from a single leaf of herbarium specimen were cut, pulverized at room temperature and soaked in a CTAB extraction solution (1% CTAB, 700 mM NaCl, 0.1 mg mL$^{-1}$ Proteinase K, 0.05 mg mL$^{-1}$ RNAse A, 0.5% N-lauroylsarcosine, 1X Tris–EDTA) under constant agitation and until tissue lysis at 56 °C (up to 6 h); an equal volume of 24:1 chloroform:isoamyl alcohol was added before centrifugation and recuperation of the aqueous phase (twice), followed by adding 7/3 volume of pure ethanol for an overnight precipitation at −20 °C. Dried pellets were resuspended in 10 mM Tris buffer and stored at −20 °C until further use. Following extraction, fragment size and concentration were controlled using TapeStation (Agilent Technologies) high sensitivity assays, according to the manufacturers' recommendations. Seven herbarium samples were converted into double-stranded libraries using the aDNA-adapted Blunt-End-Single-Tube (BEST) protocol from Carøe et al. [37]. Library preparation of the remaining six herbarium samples was outsourced to Fasteris (https://www.fasteris.com/dna/). Briefly, DNA extracts were purified using magnetic beads, optimized for the recovery of fragments of either >20 bp in the Fasteris facility for TrueSeq libraries, or >50 bp in our laboratory (Sera-mag Speed Beads) for BEST libraries. After T4-polymerase-blunting, subsequent steps included T-A ligation of adapters and 8 PCR cycles, with a final bead purification >135 bp by Fasteris for TrueSeq libraries, or used BstXI for further fill-in, ligation of adapters and 13–18 indexing PCR cycles using PFU Turbo Cx, with a final bead cleanup (2% Sera-mag Speed Beads, 18% PEG−8000, 1 M NaCl, 10 mM Tris.Cl, 0.05% Tween-20, at a Speed Beads/sample volume ratio of 1.2) for BEST libraries. PCR indexing cycles of both protocols used U-compatible polymerases. Blank controls were verified with a specific qPCR[68] at different library construction stages, and sequencing of herbarium samples was performed in four batches, each including one non-host sample (*Coffea*) as negative control, in a paired-end 2 × 150 cycles configuration on a NextSeq flow cell.

### Modern bacterial strains culture, DNA extraction and library preparation

Fifty-seven bacterial strains isolated between 1963 and 2008, mainly from Asia (Supplementary Data S1) and stored as lyophiles at −80 °C, were chosen to complement the collection of available modern genomes. Strains were grown at 28 °C on YPGA (7 g L$^{-1}$ yeast extract, 7 g L$^{-1}$ peptone, 7 g L$^{-1}$ glucose, 18 g L$^{-1}$ agar, supplemented by 20 mg L$^{-1}$ propiconazole, pH 7.2). Single cultures were used for DNA extraction using the Wizard® genomic DNA purification kit (Promega) following the manufacturer's instructions. Quality assessment was realized for concentration using QuBit (Invitrogen) and Nanodrop (Thermo Fisher Scientific) fluorometers. Library preparation of the modern strains was outsourced to Fasteris where classic TruSeq Nano DNA protocol following Nextera enzymatic DNA fragmentation was applied (Illumina). Sequencing for both historical and modern DNA was performed in a paired-end 2 × 150 cycles configuration on a NextSeq500 machine in several batches, with samples from both types of libraries being independently treated.

### Initial reads trimming and merging

Artefactual homopolymer sequences were removed from libraries when presenting entropy inferior than 0.6 using BBDuk from BBMap 37.92[69]. Adapters were trimmed using the Illuminaclip option from Trimmomatic 0.36[70]. Such reads were processed into the *post-mortem* DNA damage assessment pipeline detailed in the section below. Additional quality trimming was realized with Trimmomatic based on base quality (LEADING:15; TRAILING:15; SLIDINGWINDOW:5:15) and read length (MINLEN:30). Paired reads were merged using Adapter-Removal 2.2.2[71] with default options and all subsequent analyses were performed on both the merged and the remaining non-merged reads.

### Ancient DNA damage assessment and statistical analyses

*Post-mortem* DNA damage was measured by DNA fragment length distribution and terminal deamination patterns using mapDamage 2.2.1[41]. Alignments required for mapDamage were performed with an aligner adapted to short reads BWA-aln 0.7.15 (default options, seed disabled)[72] for the herbarium samples and Bowtie 2 (options–non-deterministic–very-sensitive)[73] for modern strains, using *Xci* reference strain IAPAR 306 genome (chromosome NC_003919.1, plasmids pXAC33 NC_003921.3 and pXAC64 NC_003922.1 of length 5,175,554 bp, 33,700 bp and 64,920 bp, respectively). MarkDuplicates in picardtools 2.7.0[74] was run to remove PCR duplicates. For each sample, reads were grouped in nucleotide length classes of 25 nucleotide-long intervals, from 15 to 290 nucleotides (i.e., 11 classes). Analysis of variance aov function ("stats" R package) was used to test the effect of protocol (BEST or TruSeq), DNA type (plasmid or chromosome), and age (years) on nucleotide length classes. When significant, a *Student t.test* was performed. Deamination rate data of each sample took into account: the number of terminal 3′ G > A substitutions, divided by the total number of reads bearing a G at the same position (according to the reference sequence). Effect of protocol (BEST or TrueSeqNano) and age (in years) on deamination rate was assessed using a glm model ("stats" R package), under quasi-binomial distribution. Effect of DNA type (plasmid or chromosome) was specifically tested using glmmPQL function, allowing the analysis of paired variables.

### Genome reconstruction

Genomes were reconstructed by mapping quality-trimmed reads to *Xci* reference strain IAPAR 306 genome using BWA-aln for short reads[72] and Bowtie 2 for longer reads[73], as defined above. Sequencing depths were computed using BEDTools genomecov 2.24.0[75]. For herbarium specimens, binary alignment map (BAM) files were extremity trimmed on their five external nucleotides at each end using BamUtil 1.0.14[76]. Single nucleotide polymorphisms (SNPs) were called with GATK v4.2 UnifiedGenotyper[77]; they were considered dubious and filtered out if they met at least one of the following conditions: "depth < 20", "minor allelic frequency < 0.9" and "mapping quality < 30". Consensus sequences were then reconstructed by introducing the high-quality SNPs in the *Xci* reference genome and replacing dubious SNPs and non-covered sites (depth = 0) by an *N*.

### Phylogeny and tree-calibration

A dataset of 171 modern *Xci* genomes (date range: 1948–2017) representative of *Xci* global diversity was built from 114 previously published genomes and 57 new genomes generated within the course of this study (Supplementary Data S1). An alignment of the 13 historical chromosome sequences with the 171 modern sequences was constructed for phylogenetic analyses, with strains of *Xanthomonas axonopodis* pv. *vasculorum* NCPPB-796 from Mauritius (isolated in 1960, GCF_013177355.1), *Xanthomonas citri* pv. *cajani* LMG558 from India (1950, GCF_002019105.1) and *Xanthomonas citri* pv. *clitoriae* LMG9045 from India (1974, GCA_002019345.1) as outgroups. Variants from modern strains were independently called and filtered using the same

parameters as for historical genomes. Recombinant regions were identified inside the *Xci* dataset (ingroup only) with ClonalFrameML[78] and removed, to minimize production of incongruent trees due to recombination during phylogenetic reconstruction. Two single nucleotide polymorphism (SNP) datasets were constructed, either within the *Xci* ingroup only, or from across the whole dataset (ingroup + outgroups). A maximum likelihood (ML) tree was constructed on both SNP alignments using RAxML 8.2.4[43] using a rapid Bootstrap analysis, a General Time-Reversible model of evolution following a Γ distribution with four rate categories (GTRGAMMA) and 1000 alternative runs[79].

As a requirement to build tip-calibrated phylogenies, the existence of a temporal signal was investigated using three different tests. First, a linear regression test between sample age and root-to-tip distances was computed at each internal node of the ML tree using Phylostems[80]. Temporal signal was considered present at nodes displaying a significant positive correlation. Secondly, a date-randomization test (DRT)[81] was performed with 20 independent date-randomized datasets generated using the R package "TipDatingBeast"[82]. Temporal signal was considered present when there was no overlap between the inferred root height 95% highest posterior density (95% HPD) of the initial dataset and that of 20 date-randomized datasets. Finally, a Mantel test with 1000 date-randomized iterations investigating whether closely related sequences were more likely to have been sampled at similar times was also performed to ensure no confounding effect between temporal and genetic structure, as recent work suggested that temporal signal investigation through root to-tip-regression and DRT could be misled in such a case[83].

Tip-dating calibration Bayesian inferences (BI) were performed on the primary SNP alignment (*Xci* ingroup) with BEAST 1.8.4[84]. Hence, the xml input file was manually edited with the number on invariant A,T,C,Gs to correct rates for the fact that SNPs only were being used, as advised by Rieux et al. [13]. Leaf heights were constrained to be equal to sample ages. Flat priors (i.e., uniform distributions) for the substitution rate ($10^{-12}$–$10^{-2}$ substitutions/site/year) and for the age of all internal nodes in the tree were applied. We also considered a GTR substitution model with a Γ distribution and invariant sites (GTR + G + I), an uncorrelated relaxed log-normal clock to account for variations between lineages, and a tree prior for demography of coalescent extended Bayesian skyline. The Bayesian topology was conjointly estimated with all other parameters during the Markov chain Monte-Carlo (MCMC) and no prior information from the tree was incorporated in BEAST. Five independent chains were run for 200 million steps and sampled every 20,000 steps, discarding the first 20,000 steps as burn-in. Broad-platform evolutionary analysis general likelihood evaluator (BEAGLE) library was used to improve computational speed[85,86]. Convergence to the stationary, sufficient sampling (effective sample size > 200) and mixing were checked by inspecting posterior samples with Tracer 1.7.1[87]. Final parameters estimation was based on the combination of the different chains. Maximum clade credibility method in TreeAnnotator[84] was used to determine the best-supported tree of the combined chains.

Rate-dating calibration outside the *Xci* clade was performed on the secondary SNP alignment (ingroup + outgroups) with BEAST 1.8.4[84]. Instead of using tip-dates, we applied a prior on the substitution rate by drawing values from a normal distribution with mean and standard deviation values fixed as those inferred using tip-dating calibration within *Xci* ingroup. All other parameters were applied as described previously.

### Phylogeography and ancestral location state reconstruction

The presence of geographic structure in the ML tree was measured through calculation of the association index (AI) and the comparison of its value with the ones computed from 1000 location-randomized

trees[88]. Non-random association between phylogeny and location was assumed when less than 5% of AI values computed from the randomized trees were smaller than the AI value of the real ML tree.

Ancestral location state was reconstructed using BEAST 1.8.4 under the same parameters as tip-dating calibration but adding a partition for location character. We modeled discrete location transitioning between areas throughout *Xci* phylogenetic history using a continuous-time Markov chain (CTMC) process under an asymmetric substitution model with a Bayesian stochastic search variable selection (BSSVS) procedure. States were recorded from countries to greater areas: East Africa (Ethiopia), West Africa (Mali and Senegal), the Caribbean (Martinique (France)), North America (United States of America), South America (Argentina and Brazil), East Asia 1 (China and Taiwan), East Asia 2 (Japan), South East Asia 1 (Cambodia, Malaysia, Myanmar, Thailand and Vietnam), South East Asia 2 (Indonesia and Philippines), South Asia 1 (Bangladesh, India and Nepal), South Asia 2 (Iran and Pakistan), West Asia (Oman and Saudi Arabia), Oceania and Pacific (Fiji, Guam, New Zealand and Papua New Guinea), North Indian Ocean islands (Maldives and Seychelles), and South West Indian Ocean islands (Comoros, Mauritius and Rodrigues, Mayotte and La Réunion (France)).

### Pathogenicity-associated genes content analysis

The presence of pathogenicity-associated genes was investigated using a list of 66 type III effectors (T3E) found in *Xanthomonas*[58,89] as well as 24 genes involved in type III secretion system (T3SS)[90] and 54 genes more distantly involved in pathogenicity (Supplementary Data 2). Alignments were performed simultaneously on all sequences either with BWA-aln or Bowtie 2 (same conditions as above), for herbarium samples and modern strains, respectively. The sequences used to assess homology were the reference strain IAPAR 306 CDS (coding sequences) when available; when not, variants from other *Xci* strains or other *Xanthomonas* CDS were used. Depth (average number of mapped reads at each base of the reference genome) was recovered using BEDTools genomecov 2.24.0[75] and coverage (proportion of reference genome covered at 1X depth) were calculated with R. Genes were considered present if their sequence was covered on more than 75% of its length. The pathogenicity-associated genes content was then projected on BI phylogenetic tree using the gheatmap function of R "ggtree" package[91].

### Reporting summary

Further information on research design is available in the Nature Portfolio Reporting Summary linked to this article.

## Data availability

The authors confirm that all data used in this study are fully available without restriction. Raw reads and consensus genomes were deposited to the NCBI Sequence Read Archive (SRA) and GenBank, respectively, under accession numbers listed in Supplementary Data S1. Accession numbers of any previously published data used in this study are also listed in Supplementary Data S1. Source data are provided with this paper.

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

## Acknowledgements

We are grateful to P. Lefeuvre, D. Richard, F. Balloux, D. Fargette, V. Llaurens, R. Debruyne, and Á. Pérez-Quintero for valuable comments and discussions. We thank L. Bui Thi Ngoc (SOFRI, Viet Nam), B. Canteros (INTA, Argentina), B. Carter (FERA, UK), R. Davis (NASQ, Australia), A. Hamza (INRAPE, the Comoros), G. Johnson (Horticulture4Development, Australia), N. Le Mai (PPRI, Viet Nam), W. Wu (National Chung Hsing University, Taiwan) and M. Zakria (NARC, Pakistan) for providing citrus canker lesions and/or bacterial strains. We also thank Meghann Toner (collections management, Department of Botany, National Museum of Natural History, Washington, DC, USA) and Shannon Dominick (collections coordinator, US National Fungus Collections (Herbarium BPI), USDA-ARS Mycology and Nematology Genetic Diversity and Biology Laboratory, Beltsville, MD, USA) for allowing us to consult and sample historical *Citrus* specimens. Collection of any plant material used in this study complies with institutional, national, and international guidelines. Permission to collect and analyze each historical specimen included in this study was provided by the herbarium institutions (and their curators) from which they were sampled. Computational work was performed on the CIRAD-UMR AGAP HPC data center of the South Green bioinformatics platform (http://www.southgreen.fr/) and MESO@LR-Platform at the University of Montpellier (https://hal.umontpellier.fr/MESO). Laboratory work was conducted on the Plant Protection Platform (3P, IBISA). This work was financially supported by Agence Nationale de la Recherche (JCJC MUSEOBACT contrat ANR-17-CE35-0009-01 received by A.R., N.B., O.P., and L.G.), the European Regional Development Fund (ERDF contract GURDT I2016-1731-0006632 for O.P., K.B., N.B., and A.R.), Région Réunion, the Agropolis Foundation (Labex Agro—Montpellier, E-SPACE project number 1504-004 for O.P., L.G., and A.R., MUSEOVIR project number 1600-004 for L.G. and A.R.), the SYNTHESYS Project http://www.synthesys.info/ financed by European Community Research Infrastructure Action under the FP7 "Capacities" Program (grants GB-TAF-6437 for A.R. and GB-TAF-7130 for L.G.) & CIRAD/AI-CRESI- 3/2016 for L.G. and A.R. The Ph.D. of P.E.C. was co-funded by ED 227, Muséum national d'Histoire naturelle and Sorbonne Université, Ministère de l'Enseignement Supérieur, de la Recherche et de l'Innovation.

## Author contributions

All authors participated in scientific discussion of the manuscript and provided comments, critiques, and/or approvals prior to submissions and revisions. A.R., L.G., N.B., P.E.C., and O.P. conceptualized the research project. M.G., C.B., and T.M.A.U. provided access to the historical material to L.G., A.R., P.E.C., and N.B. O.P. managed the production of the modern genomes from our bacterial collection. K.B., P.E.C., N.B., and A.R. performed molecular lab work. P.E.C. curated the dataset. F.C. and T.T.C. provided advices for statistical analyses. P.E.C. performed all the genetic analyses under the supervision of A.R., L.G., and N.B. P.E.C., A.R., N.B., and L.G. wrote the first draft and all authors contributed to the final version.

## Competing interests

The authors declare no competing interests.
