## [Peer Review File · Nature Communications]

Herbarium specimen sequencing allows precise dating of *Xanthomonas citri* pv. *citri* diversification historyReviewer #1 (Remarks to the Author):

The authors used an extensive sampling covering the last 70 years of Xci strains evolution, along with a broad representation of Citrus specimens in herbaria dating back to as early as 1845 (impressive). They sequenced historical bacterial genomes of Xci from 13 herbarium samples showing typical canker symptoms. They compared the historical genomes to those of 171 modern strains representative of the worldwide genetic diversity. The study is exciting and a pleasure to read. The experimental design is appropriate.

Main suggestions:

The authors justified the study using: two dating methodologies known to yield potential misleading estimates (Ho et al. 2008, Rieux et al. 2014). This needs to be discussed whether Patané et al. 2019 did generate any misleading information as the authors claimed. The authors processed reads on reference strain IAPAR 306 genome. Considering the complicated relationship between A, A*, and Aw, please also process the reads using A* and Aw as reference to see whether there are any significant differences in term of % of mapped reads. "The chromosome sequences displayed a coverage (proportion of reference genome covered) at 1X between 94.6 and 98.2%, with mean depth (average number of mapped reads at each base of the reference genome) of 6.2 (HERB_1937) to 96.2X (HERB_1922) (Table 2). Lower coverages (49.7 to 97.1%) but higher mean depths (17.9 to 130.3X) were obtained for 275 reads mapping to plasmid references (S2 Table)." The description here is confusing to me. Can the author draw a figure to show the coverage and depth of the chromosome? It is hard to interpret the quality based on the description here. To make it easy to follow, the authors can compare that with the genome sequencing data of the modern strains. I totally understand the challenges of sequencing ancient DNA. It is ok that ancient genomic DNA has lower quality than modern strains. But the information is useful to interpret the data.

Fig. 1A. Please include some modern strains for comparison purpose.

significantly lower deamination rates were observed for BEST protocol than TruSeq. The observation was based on the protocols to be used on non-overlapping samples. If ancient DNA samples are available, please sequence the DNA using both protocols for the same DNA samples. Please also discuss whether deamination rates in ancient DNA are an artificial effect based on the observation here. Based on the data, the discussion on deamination rates (lines 411-422) might need to be revised.

For the samples collected in different time, does the storage length affects the deamination rates? For the analysis of SNPs, it is unclear to me whether the authors excluded the nucleotides from low coverage regions in ancient samples.

Fig 3. Please include the strains from South America and USA for this analysis.

In the pathogenicity genes, please specifically report PthA1, 2, 3, and 4 owing to their importance including their presence/absence and changes in the central repeat region. Please also discuss their evolution by comparing the ancient strains and modern strains.

The discussion including some analyses related to pathogenicity genes that can be moved to results.

I suggest to focus the discussion on main findings of the study. The main findings seems to be buried in the too lengthy discussion.

Reviewer #2 (Remarks to the Author):

Campos et al. have sequenced and analyzed 13 herbarium strains of *Xanthomonas citri* subsp. *citri*, with one strain dating as far back as 1845. This allowed them to obtain more precise dates for Xci evolutionary history than previously. The methodology used is state of the art, and conclusions drawn on the whole are sound. Therefore this is an important contribution to the understanding of Xci evolutionary history and lays the groundwork for similar future studies that could include the sequencing and analysis of herbarium strains of other plant pathogens.

My main concern was the fact that the authors did not assemble the 57 genomes sequenced for this study, relying instead on genome reference mapping. The authors acknowledge this as limitation. But de novo assembly of 57 genomes, in this day and age, is no big deal, and would not

have required excessive time nor resources. In any case, I believe the results should not be significantly compromised by the lack of this de novo assembly, so I don't see this as a requirement for publication. It can be seen simply as a quirk of the study.

There follow specific comments, most of them minor.

Title: The word Datation in the title seems to be French; in English the expected word is Dating.

When referring to tables and figures, it's customary to say Table X and Figure Y, and not X Table and Y Figure.

L99: ...57 of which *were* specifically sequenced...

L120:while HERB_1937 *has been* processed...

L177-178: Were there any checks for continuous overlap of reads within large reconstructed regions mapped to X. citri 306, to further minimize the possibility of assembly error in those regions? No need to be a thorough analysis, but it should be done at least for a few different regions within each of the herbarium genomes.

L188-189: suggestion: ... to *minimize* production of incongruent trees *due to recombination* during phylogenetic reconstruction.

L191-193: Was SNP ascertainment correction employed? Otherwise branch lengths, and possibly some parts of the ML tree, may be biased; this may be particularly relevant at short internodes (such as A/A*/Aw divergences), where high bootstraps may further amplify the signal of a (potentially) wrong phylogenetic relationship. Furthermore (if ascertainment correction was not employed), the potentially incorrect ML branch lengths may bias tip-dating tests for temporal signal.

L208: Do authors mean "Leaf heights were constrained to be *equal* to sample ages", otherwise please clarify.

L236: It is not clear why the authors used an asymmetric substitution model between areas, especially given that X. citri is easily dispersed (e.g., by rain and market, even in the past). Why not a symmetric model? Did the authors test and compare both models in some way?

L245-255: Did the authors check for continuous overlap of reads within such reconstructed genes (or gene segments)?

L267: HERB_1937 is clearly an outlier in terms of endogenous DNA and depth. Since it is the only one from a previous study, can the authors point to some methodological difference between the previous study and this would that would explain these low values?

L268: The sentence that starts with "Respective values" is not clear. What are these values? what do the authors means by "control"? it seems that many important details are missing.

L316-317: Refer to comment for lines "191-193" above.

L326-327: Refer to comment for lines "191-193" above.

L331-332: "... no overlap of the 95% HPD for the *root* "

L344-351: Were rates corrected for the fact that SNPs only were being used (hence a much higher rate per site, compared to genome-wide, which include a much larger amount of invariable sites)? If not, posterior time distributions may be biased (due to their inverse relationship). Furthermore, there is the possibility of high support for a wrong clade (as mentioned in comment for lines "191-193"), which could also confound which lineage appeared first.

L346: suggestion: Although it is reasonable to report a node's mean posterior time, a more conservative suggestion would be: "...was inferred to date to at least 1,437 AD, possibly being even older, at most 962 AD."

L433: Not "somehow" but "somewhat"

L455-462: Besides the difference in dating methodology, also check the possibilities mentioned in comment for lines "344-351" above, so that priors on rates can be taken to be reasonably meaningful.

L507-510: Correct, but state that in such cases a threshold on 95% or 99% of a large set of individuals of a given lineage carrying the paralog can be used as a surrogate for core effectome, which is also within current usage trends.

L513-515: The observations about xopC1 and XAC1496 were noted in reference 34.

L531: Not "representants" but "representatives"

Reviewer #3 (Remarks to the Author):

The manuscript by Campos et al. presents a dated phylogeny of the crop pathogen *Xanthomonas citri* pv. *citri* (Xci), including 13 genomes reconstructed from historic herbarium specimens. The study is an important contribution as it could identify the likely place and time of origin as well as the diversification of Xci. This advances the understanding of pathogen diversification. The study also shows that herbarium specimens can be a useful resource to obtain sequences from bacterial plant pathogens.

The manuscript is clearly written, and the figures are well presented and clear.

Below are some specific comments that I hope are useful for revising the manuscript.

l. 76: maybe consider adding the common name for these species as you have it for the ones mentioned in the next sentence

l. 77-78: Shouldn't the full Latin names be used here as it's the first occurrence of the species names?

l. 125: In addition to the *Coffea* sp. samples used as negative controls, was an extraction blank used?

l. 128: For the BEST library build, what purification method was used after the adapter fill-in (beads or columns)? Also, how was the amplification PCR performed (which polymerase, number of cycles, purification method)?

l. 129: Contamination can also happen during library prep, not only during extraction. How did you ensure that there was no contamination at Fastaris for the herbarium samples as they also did the library build for the modern samples?

l. 157: Can you add the genome size for the reference strain used here for mapping?

l. 163-164: Why was 3' G>A used and not also 5' C>T? Do you get the same results when using 5' C>T?

l. 173: Fig. 1B shows that there is still a much higher G>A rate at position 5 for herbarium samples compared to the modern samples. So, removing the first 5 bases may not be sufficient to remove aDNA damage. Have you tried to recalibrate base qualities with mapDamage or remove C>T and G>A positions from the analysis?

I. 174: What GATK version did you use?

I. 250-252: How exactly did you do the mapping to the genes? Was it done against all genes simultaneously (all genes in one reference fasta file) or one gene at a time? Also, did you filter for MAPQ when assessing if a gene was present or not to remove unapacically mapped sequences? This is especially of concern for the historical samples as they contain many short fragments.

L. 252: Please specify what you mean with coverage here as it is sometimes used as a synonym for average sequencing depth and sometimes as the % of a sequence covered.

Fig. 1A: Maybe also add the modern samples to Fig 1A to show that the fragments are much shorter in herbarium samples compared to modern samples.

Fig. 1B: Consider to also add 5' C>T to the figure.

I. 323-325: Maybe add a figure to illustrate the geographic clustering. You could e.g., give each lineage a different color and display the fraction of samples belonging to each lineage as a pie chart for each geographic region. That would make it much easier to see that e.g. A3 only occurs in South Asia 1.

Fig. 2A: What is the meaning of the different colored points in this figure? I assume the red ones are the herbarium samples and the other ones are the modern ones? Please explain this in the legend.

I. 346 (and other occurrences afterwards): Please add a unit (e.g. CE) to all your dates to avoid confusion.

I. 416-417: It is very interesting that you observe different deamination values with the different library methods. However, I'm not totally convinced that this is due to the library method itself. The methods section lacks some details about e.g. the polymerase used for the amplification step (some polymerases like AmpliTaq Gold have a bias towards shorter fragments) and purification method (beads vs. columns). And if beads were used, what bead:DNA ratio was used as this determines the size selection? In addition, the deamination rate is very sample specific and may depend on the specimen preparation and storage conditions in the herbarium. For example, Fig. 3b of Weiß et al. (2016) shows that some samples collected before 1900 can have C>T misincorporations below 2 % while others of the same age are above 4%. So to actually test differences between the two library methods, it would be better to use the same DNA extract of the same sample.

L. 418: What polymerases were actually used for the amplification step? Did you use a polymerase incapable of recovering uracils?

Reviewer #4 (Remarks to the Author):

* General Comments

Campos et al. present a thorough investigation of historical and modern genomes reconstructed from the Citrus pathogen *Xanthomonas citri* pv. *citri*. Their results are very compelling, and I am impressed by the comprehensiveness of their analyses, from ancient DNA authentication and quality control all the way to Bayesian phylogenetic and phylogeographic modeling.

I have only a small number of technical comments and minor concerns that I would like to see addressed. Overall however, I think the present study does a great job of highlighting the power and utility of herbarium genomics, especially to study plant pathogens.

** Methods

- This is more of a curiosity-driven question, but did you also map sequence reads to the

suspected hosts? If so, was the host always correctly identified in the herbarium record? (I'm not proposing the authors to do this if they haven't, but it would be interesting to report if they did already do this analysis anyways.)

- Along similar lines, I think it's a little confusing to refer to the Xci DNA content as "endogenous" (e.g. in Table 2 and the Discussion), since really, the "endogenous DNA" of the specimen would be the host plant, with Xci being a pathogen infecting it

- For merging aDNA paired-end reads, please also report the percentage of merged reads (in addition to median insert length after merging), since a low fraction of merged pairs can be indicative of long fragments present in the library which would be missed when reporting just the length statistics of the merged fraction

- Also, did you continue the data analysis with only the merged fraction of the herbarium libraries? This is unclear from the current Methods.

- picardtools MarkDuplicates doesn't properly remove duplicates for merged aDNA reads. I'm not proposing to re-run everything, but it would be a helpful sanity check to compare the percentage of duplicates removed from picardtools vs. a tool optimized for aDNA such as DeDup:

<https://github.com/apeltzer/DeDup>

- The authors use ClonalFrameML to remove recombinant regions from their dataset, as those violate the assumptions of their subsequent phylogenetic analysis. I would be interested in additional details as to how well this actually eliminates reticulations from their trees, for example by comparing phylogenetic networks before and after the ClonalFrameML-based removal (using SplitsTree, for example)

- Regarding the pathogenicity-associated gene content analysis: it appears to me that there may be quite a few homologous genes in the set of effectors you map to. It would be good to give a little more detail in the Methods section on how you handle the issues that may arise from this. Do you map to these one-by-one? or all at once? how do you deal with situations where a read may map equally well to multiple genes? Do you compare the genic coverage to a measure of whole-genome coverage of the same individual to assess potential issues with copy-number or multiple mappings?...

** Analyses

- Fig. 2A: there seems to be a mismatch between the slope reported in the figure inset vs. in the text (line 329).

- It's a little confusing to have the BEAST analysis already mentioned when describing Fig. 2B, before the analysis is properly introduced for Figure 3. Maybe this could be restructured slightly to make more clear what analysis is done for Fig. 2B?

- Table 3: it would be helpful to clarify that negative dates correspond to dates "BC" in this table (and elsewhere), to avoid confusion with dates "Before Present" that are sometimes presented in a similar fashion.

* Specific Comments

- l.72/73: Not sure it's clear to the reader what "quarantine organism status" is supposed to mean

- l.92: It would be helpful for understanding to more clearly state what the "sampling covering 70 years" refers to here (it looks like it refers to the time frame during with the "modern" samples analysed here were sampled?)

- l.320: rephrase "Lineage A2 corresponds to lineage A2" -- confusing otherwise

- l.526: you mention 163 genomes here, but previously you refer to 171+13 genomes

* Writing

- l.24: "shed light *on*..."

- l.196: "*using* three different tests..."

- l.376: "we *used/performed* a rate-dating analysis..."

- l.472: "*constrained*" instead of restrained

REVIEWER COMMENTS

Reviewer #1 (Remarks to the Author):

The authors used an extensive sampling covering the last 70 years of Xci strains evolution, along with a broad representation of Citrus specimens in herbaria dating back to as early as 1845 (impressive). They sequenced historical bacterial genomes of Xci from 13 herbarium samples showing typical canker symptoms. They compared the historical genomes to those of 171 modern strains representative of the worldwide genetic diversity. The study is exciting and a pleasure to read. The experimental design is appropriate.

Main suggestions:

- The authors justified the study using: two dating methodologies known to yield potential misleading estimates (Ho et al. 2008, Rieux et al. 2014). This needs to be discussed whether Patané et al. 2019 did generate any misleading information as the authors claimed.

>> We do not claim that Patané et al. (2019)¹ generated misleading information (which we cannot affirm), but rather discuss the fact that the discrepancy between their and our estimates probably arises from differences in the chosen dating methodologies (node & rate- vs tip- dating), with our approach being previously reported to yield more accurate and robust values (see lines 499-507)

- The authors processed reads on reference strain IAPAR 306 genome. Considering the complicated relationship between A, A*, and Aw, please also process the reads using A* and Aw as reference to see whether there are any significant differences in term of % of mapped reads.

>> Our reference-based approach used to build a SNP dataset requires the use of a single reference genome. We initially selected strain IAPAR 306 for two reasons. First it is the best annotated Xcc genome available. Second this strain belongs to pathotype A, like the vast majority (90%) of the genomes analyzed in this study. Following your comment, we mapped the reads to two other

reference genomes (strains JK2-10 & 12879 for pathotype A* & A^W, respectively). As expected, we observed some variation in % of mapped reads but importantly, mapping rates were on average always higher between the phylogenetically identified pathotype and their reference genomes (see boxplots below). Hence, we believe IAPAR 306 was the best reference sequence to use in our study.

- “The chromosome sequences displayed a coverage (proportion of reference genome covered) at 1X between 94.6 and 98.2%, with mean depth (average number of mapped reads at each base of the reference genome) of 6.2 (HERB_1937) to 96.2X (HERB_1922) (Table 2). Lower coverages (49.7 to 97.1%) but higher mean depths (17.9 to 130.3X) were obtained for reads mapping to plasmid references (S2 Table).” The description here is confusing to me. Can the author draw a figure to show the coverage and depth of the chromosome? It is hard to interpret the quality based on the description here. To make it easy to follow, the authors can compare that with the genome sequencing data of the modern strains. I totally understand the challenges of sequencing ancient DNA. It is ok that ancient genomic DNA has lower quality than modern strains. But the information is useful to interpret the data.

>> We rephrased as following: “Coverage (proportion of reference genome covered) at 1X ranged between 94.6 to 98.2% for the chromosome, and 49.7 to 97.1% for the plasmids (Table 2 & S2), due

to the fact that plasmid contents are usually highly variable². By contrast mean depths (average number of mapped reads at each base of the reference genome) were between 6.2X and 96.2X for the chromosome and between 17.9X and 130.3X for the plasmids, reflecting the presence of multiple copies of plasmids in *Xanthomonas* (Table 2 & S2).” The above comment refers to the “Historical genomes reconstruction & ancient DNA damage pattern assessment” section of the results. We believe it is not judicious to compare depths and coverages of modern vs. herbarium strains to the reference as the latter are highly dependent on the proportion of *Xci* within the total amount of DNA sequenced, necessarily lower for such samples than for DNA from freshly cultured bacteria. Nevertheless, we added a section summarizing the modern genomes reconstruction statistics in our revised manuscript (L 321-329) and provide below such comparisons for illustration purpose:

- Fig. 1A. Please include some modern strains for comparison purpose.

>> Done.

- Significantly lower deamination rates were observed for BEST protocol than TruSeq. The observation was based on the protocols to be used on non-overlapping samples. If ancient DNA samples are available, please sequence the DNA using both protocols for the same DNA samples. Please also discuss whether deamination rates in ancient DNA are an artificial effect based on the observation here. Based on the data, the discussion on deamination rates (lines 411-422) might need to be revised. For the samples collected in different time, does the storage length affects the deamination rates?

The deamination rates, assessed in our study for 13 herbarium samples, were a prior for ancient DNA authentication. Collection time (*i.e.* storage length) has been suggested to influence deamination rates, as shown for more than 50 herbarium samples over a 1750-2000 time span³, so we examined this possibility. Fig S2 shows a first “protocol effect” on deamination rates ($p < 0.0001$), which had to be taken into account. Thus, we performed our analysis within each protocol subgroup, covering similar timespans (7 samples collected from 1845 to 1974, versus 6 samples collected from 1852 to 1963), and observed a significant, time-dependent, effect on terminal deamination rates ($p < 0.0001$).

Experimental validation of the protocol effect was, despite its interest, neither the aim of this study, nor possible, due to the scarcity of symptom-harboring plant material (now stated in results section, lines 309-310). For each of the 13 samples, the totality of the material was either sent to Fasteris (a biotechnology company providing sequencing services), or processed in our laboratory, for bead purification prior to library conversion. We thus shortened our discussion (lines 463-467), removing any “protocol effect” interpretation.

After supplementary discussions with Fasteris, we now list in more detail the steps which are significantly different between each protocol (Materials and Methods). We believe that the size of the DNA fragments may be linked to deamination, for three reasons: i) The Truseq protocol includes two specific bead purification steps, in favor of shorter fragments than the ones used in the BEST protocol (when comparing fragment length categories, the BEST library was enriched in longer fragments (66-140 nt), while the TruSeq Nano library was enriched in smaller fragments (15-40 nt and 41-65 nt) (p -value < 0.0001 in all cases, aov function, R package, data not shown); ii) the size of the merged sequences obtained is significantly smaller for the “Truseq” protocol; iii) Gutaker et al. (2017)⁴ showed, comparing two different DNA binding methods on same samples, that a decrease in median fragment length was accompanied by an increase in C-to-T substitutions.

- For the analysis of SNPs, it is unclear to me whether the authors excluded the nucleotides from low coverage regions in ancient samples.

>> SNPs were filtered out at positions where depth $< 20X$ both for modern and ancient genomes, except for HERB_1937 (the outlier in terms of sequencing depth) for which threshold was fixed to depth $< \text{average depth} + 1\text{sd}$ (*i.e.* $9X$), as described in a previously published study⁵.

- Fig 3. Please include the strains from South America and USA for this analysis.

>> Fig 3 and associated analyses were initially made using all strains listed in Table S1 and thus already includes 14 South American and 2 USA strains, respectively.

- In the pathogenicity genes, please specifically report PthA1, 2, 3, and 4 owing to their importance including their presence/absence and changes in the central repeat region. Please also discuss their evolution by comparing the ancient strains and modern strains.

The discussion including some analyses related to pathogenicity genes that can be moved to results.

>> Done. Specific information about PthA genes was added in the results section (L433-440) and descriptive parts of the discussion on effectors were moved to results (L446-452)

- I suggest to focus the discussion on main findings of the study. The main findings seems to be buried in the too lengthy discussion.

>> As mentioned above and below the discussion was simplified, elements were moved to the results section and discussion related to the core effectome was removed.

Reviewer #2 (Remarks to the Author):

Campos et al. have sequenced and analyzed 13 herbarium strains of *Xanthomonas citri* subsp. *citri*, with one strain dating as far back as 1845. This allowed them to obtain more precise dates for Xci evolutionary history than previously. The methodology used is state of the art, and conclusions drawn on the whole are sound. Therefore this is an important contribution to the understanding of Xci evolutionary history and lays the groundwork for similar future studies that could include the sequencing and analysis of herbarium strains of other plant pathogens.

- My main concern was the fact that the authors did not assemble the 57 genomes sequenced for this study, relying instead on genome reference mapping. The authors acknowledge this as limitation. But *de novo* assembly of 57 genomes, in this day and age, is no big deal, and would not have required excessive time nor resources. In any case, I believe the results should not be significantly compromised by the lack of this *de novo* assembly, so I don't see this as a requirement for publication. It can be seen simply as a quirk of the study.

>> *De novo* assembly of historical genomes was tricky due to small degraded DNA fragments and mixed taxonomic origins. As such we decided to treat all historical and modern genomes the same way using reference-mapping approaches. For this reason, we did not perform any *de novo* assembly on modern genomes either but instead provide access to raw reads to anyone interested in such data.

There follow specific comments, most of them minor.

- Title: The word Datation in the title seems to be French; in English the expected word is Dating.

>> We modified it accordingly.

- When referring to tables and figures, it's customary to say Table X and Figure Y, and not X Table and Y Figure.

>> We modified it accordingly.

- L99: ...57 of which *were* specifically sequenced...

>> Done.

- L120:while HERB_1937 *has been* processed...

>> Done.

- L177-178: Were there any checks for continuous overlap of reads within large reconstructed regions mapped to *X. ctiri* 306, to further minimize the possibility of assembly error in those regions? No need to be a thorough analysis, but it should be done at least for a few different regions within each of the herbarium genomes.

>> Yes, most regions displayed overlapping reads ensuring the right order of genomic regions (e.g. region from 30 to 330 bp on *Xci* chromosome displayed below).

However there were drops of depth reaching 0X (see whole chromosome sequence below, red arrows highlighting such regions for at least one historical sample) that imply an impossibility to ascertain genomic architecture of reconstructed sequences of historical samples, on which *de novo* assembly could not be performed. Nonetheless, the genomic architecture of historical, as well as modern, genomes was not essential in our study and no further checks were performed.

- L188-189: suggestion: ... to *minimize* production of incongruent trees *due to recombination* during phylogenetic reconstruction.

>> We modified accordingly.

- L191-193: Was SNP ascertainment correction employed? Otherwise branch lengths, and possibly some parts of the ML tree, may be biased; this may be particularly relevant at short internodes (such as A/A*/Aw divergences), where high bootstraps may further amplify the signal of a (potentially) wrong phylogenetic relationship. Furthermore (if ascertainment correction was not employed), the potentially incorrect ML branch lengths may bias tip-dating tests for temporal signal.

>> Ascertainment correction was not employed as we do not believe the way SNP data was generated requires it. Ascertainment bias may arise when data have not been obtained randomly with respect to the observed data patterns⁶. For example, SNPs might have first been identified from a small panel (e.g. one specific pathotype, or one species only in our case) before being typed in a larger sample of individuals (e.g. several pathotypes or species, respectively). However, for both SNP datasets generated in our study either including or not outgroup sequences, variants were directly and independently identified on the two multi-genome alignments, a procedure which should not lead to any ascertainment bias.

- L208: Do authors mean "Leaf heights were constrained to be *equal* to sample ages", otherwise please clarify.

>> Yes, we rephrased accordingly.

- L236: It is not clear why the authors used an asymmetric substitution model between areas, especially given that *X. citri* is easily dispersed (e.g., by rain and market, even in the past). Why not a symmetric model? Did the authors test and compare both models in some way?

>> It is demonstrated that Citrus canker is disseminated on long distances by plant material transport⁷. The history of Citrus consists mostly of a East to West general movement (origin in Asia 8 Mya, domestication in China at least 4,000 ya, initial expansion in the XIIth century then a more intensive expansion linked to colonization starting in the XVth century and compounded by grafting methods⁸) and supports a global asymmetrical dissemination of the disease, from infected to non-infected areas. Upon the disease becoming global, dissemination may have become symmetric. As asymmetric models do not exclude symmetric scenarios (as particular cases) we kept an asymmetric model for our phylodynamic reconstruction. Interestingly, when looking at the rate transition matrix, we observed values that are both symmetric and asymmetric, comforting us in our choice. In addition, we re-ran our analysis using a symmetric model which did not significantly affect the outcome of the ancestral state reconstruction locations.

- L245-255: Did the authors check for continuous overlap of reads within such reconstructed genes (or gene segments)?

>> Reads did not necessarily overlap over the totality of the target ORF. We chose a threshold of 75% of their sequence length as a criterion for presence, as described in M&M. Visual checks were realized on all alignment for overlaps of reads along the sequences of genes identified as varyingly present. ORFs covered at 100% all had overlaps of reads along their sequences.

- L267: HERB_1937 is clearly an outlier in terms of endogenous DNA and depth. Since it is the only one from a previous study, can the authors point to some methodological difference between the previous study and this would that would explain these low values?

>> Not really. HERB_1974 was actually sampled at the same time and from the same herbarium as HERB_1937, and was also treated using an identical methodology. We believe that differences in % of endogenous DNA and depth could rather arise from contrasted infection severities at sampling times leading to bacterial density variability in plant tissues, as often seen in modern samples.

- L268: The sentence that starts with "Respective values" is not clear. What are these values? what do the authors means by "control"? it seems that many important details are missing.

>> Details are now included in the Materials and Methods (lines 134 to 137). Three series of extractions were processed in this study. One non-host (*Coffea* sp.) control was used per extraction series: the aim of this control was to obtain endogenous plant DNA, processed exactly like the other samples, but expected to contain no *Xci* nor *Citrus* DNA. The previously displayed values were confusing (as the low number of reads mapping to the *Xci* reference could be conserved among other bacteria) and hence removed from our revised manuscript. Instead, using a blast-based approach, we now confirmed that none of the non-host controls contained any *Citrus* nor *Xci*-specific DNA fragments, thus ruling out in-lab contamination (lines 277 to 279). Blank extractions (sample buffers only) were used as well, and checked to test negative with specific qPCRs at different library

construction steps. These blank controls were not included in the sequencing: adding such a control, with as little as no DNA, would have disequibrated the sequencing process within the flow cell (sequencing platform recommendations).

- L316-317: Refer to comment for lines "191-193" above.

>> See our response to the above comment.

- L326-327: Refer to comment for lines "191-193" above.

>> See our response to the above comment.

- L331-332: "... no overlap of the 95% HPD for the *root* "

>> We rephrased accordingly.

- L344-351: Were rates corrected for the fact that SNPs only were being used (hence a much higher rate per site, compared to genome-wide, which include a much larger amount of invariable sites)? If not, posterior time distributions may be biased (due to their inverse relationship). Furthermore, there is the possibility of high support for a wrong clade (as mentioned in comment for lines "191-193"), which could also confound which lineage appeared first.

>> Yes, rates were corrected for the fact that SNPs only were loaded in BEAST by manually editing the xml with the number of invariants A,T,C & Gs, as described in Rieux A et al (2016) - Supplementals 1a⁹. We added this information in our revised manuscript (line 215-217).

- L346: suggestion: Although it is reasonable to report a node's mean posterior time, a more conservative suggestion would be: "...was inferred to date to at least 1,437 AD, possibly being even older, at most 962 AD."

>> Sentence now reads: "...was inferred to date to at least 1437 CE, possibly being even older, at most 962 CE (95% HPD), with an average around 1218 CE.)

- L433: Not "somehow" but "somewhat"

>> Done.

- L455-462: Besides the difference in dating methodology, also check the possibilities mentioned in comment for lines "344-351" above, so that priors on rates can be taken to be reasonably meaningful.

>> See our response to the above comment.

- L507-510: Correct, but state that in such cases a threshold on 95% or 99% of a large set of individuals of a given lineage carrying the paralog can be used as a surrogate for core effectome, which is also within current usage trends.

>> Discussion about core effectome was removed from the discussion as this was not an objective of our study and the representation of outgroup strains is not suitable for such comparison.

- L513-515: The observations about xopC1 and XAC1496 were noted in reference 34.

>> This information has been corrected. The corresponding paragraph is now in the Results section.

- L531: Not “representants” but “representatives”

>> Done.

Reviewer #3 (Remarks to the Author):

The manuscript by Campos et al. presents a dated phylogeny of the crop pathogen *Xanthomonas citri* pv. *citri* (Xci), including 13 genomes reconstructed from historic herbarium specimens. The study is an important contribution as it could identify the likely place and time of origin as well as the diversification of Xci. This advances the understanding of pathogen diversification. The study also shows that herbarium specimens can be a useful resource to obtain sequences from bacterial plant pathogens.

The manuscript is clearly written, and the figures are well presented and clear.

Below are some specific comments that I hope are useful for revising the manuscript.

- l. 76: maybe consider adding the common name for these species as you have it for the ones mentioned in the next sentence

>> Done.

- l. 77-78: Shouldn't the full Latin names be used here as it's the first occurrence of the species names?

>> Done.

- l. 125: In addition to the *Coffea* sp. samples used as negative controls, was an extraction blank used?

>> Yes, blank extractions (sample buffers only) were used as well, and checked to test negative with specific qPCRs at different library construction steps. These blank controls were not included in the sequencing: adding such a control, with as little as no DNA, would have disequibrated the sequencing process within the flow cell (sequencing platform recommendations), as now stated in our revised manuscript (Lines 120-121).

- I. 128: For the BEST library build, what purification method was used after the adapter fill-in (beads or columns)? Also, how was the amplification PCR performed (which polymerase, number of cycles, purification method)?

Materials and methods (lines 126-137) now includes these details and reads “Briefly, DNA extracts were purified using magnetic beads, optimized for the recovery of fragments of either >20 bp in the Fasteris facility for TrueSeq libraries, or >50 bp in our laboratory (Sera-mag Speed Beads) for BEST libraries. After T4-polymerase-blunting, subsequent steps included T-A ligation of adapters and 8 PCR cycles, with a final bead purification >135 bp by Fasteris for TrueSeq libraries, or used BstXI for further fill-in, ligation of adapters and 13 to 18 indexing PCR cycles using PFU Turbo Cx, with a final bead cleanup (2% Sera-mag Speed Beads, 18% PEG-8000, 1M NaCl, 10mM Tris.Cl, 0.05% Tween-20, at a Speed Beads/sample volume ratio of 1.2) for BEST libraries. PCR indexing cycles of both protocols used U-compatible polymerases. Blank controls were verified with a specific qPCR¹⁰ at different library construction stages, and sequencing of herbarium samples was performed in four batches, each including one non-host sample (*Coffea*) as negative control, in a paired-end 2x150 cycles configuration on a NextSeq flow cell.”

- I. 129: Contamination can also happen during library prep, not only during extraction. How did you ensure that there was no contamination at Fasteris for the herbarium samples as they also did the library build for the modern samples?

>> Between ancient samples and modern samples, the following precautions were employed to avoid contamination : *i*) for BEST libraries, the use of a specific laboratory, where no modern samples have ever been processed; *ii*) for Truseq libraries, the use of significantly distant time periods (several months apart) between the processing of ancient and modern samples. Furthermore, *i*) the usage of a specific and different index for each sample, at both ends for ancient DNA samples and at only one end for modern samples, and *ii*) non-host controls (*Coffea* samples) processed at each batch of samples, allowed a regular control of the absence of any detectable contamination.

- I. 157: Can you add the genome size for the reference strain used here for mapping?

>> Done.

- I. 163-164: Why was 3' G>A used and not also 5' C>T? Do you get the same results when using 5' C>T?

> Yes, same results were obtained when looking at 5' C>T substitutions as now represented Fig 1, Fig S1, and stated lines 293 and 313.

- I. 173: Fig. 1B shows that there is still a much higher G>A rate at position 5 for herbarium samples compared to the modern samples. So, removing the first 5 bases may not be sufficient to remove aDNA damage. Have you tried to recalibrate base qualities with mapDamage or remove C>T and G>A positions from the analysis?

>> Yes, we have tried the “recalibrate base qualities” option in mapDamage and ended up with the exact same SNPs as when this option was disabled (after applying our various quality filters described lines 182-185).

- I. 174: What GATK version did you use?

>> We used GATK 4.2, as now specified line 181.

- I. 250-252: How exactly did you do the mapping to the genes? Was it done against all genes simultaneously (all genes in one reference fasta file) or one gene at a time? Also, did you filter for MAPQ when assessing if a gene was present or not to remove unapacifically mapped sequences? This is especially of concern for the historical samples as they contain many short fragments.

>> All genes were combined in a single fasta file to allow for a simultaneous mapping of reads against each sequence under options promoting high mapping specificity. The reads were not subsequently filtered for MAPQ but BAM files were visually checked for historical samples, and for modern ones when relevant.

- L. 252: Please specify what you mean with coverage here as it is sometimes used as a synonym for average sequencing depth and sometimes as the % of a sequence covered.

>> Done (lines 260 & 262).

- Fig. 1A: Maybe also add the modern samples to Fig 1A to show that the fragments are much shorter in herbarium samples compared to modern samples.

>> Done.

- Fig. 1B: Consider to also add 5' C>T to the figure.

>> Done.

- I. 323-325: Maybe add a figure to illustrate the geographic clustering. You could e.g., give each lineage a different color and display the fraction of samples belonging to each lineage as a pie chart for each geographic region. That would make it much easier to see that e.g. A3 only occurs in South Asia 1.

>> Done, see new Figure S6.

- Fig. 2A: What is the meaning of the different colored points in this figure? I assume the red ones are the herbarium samples and the other ones are the modern ones? Please explain this in the legend.

>> Done: Root-to-tip regression lines plotted either in red solid line when integrating historical genomes (red dots), and in blue dashed line when only computed from modern genomes (black dots).

- I. 346 (and other occurrences afterwards): Please add a unit (e.g. CE) to all your dates to avoid confusion.

>> Done.

- I. 416-417: It is very interesting that you observe different deamination values with the different library methods. However, I'm not totally convinced that this is due to the library method itself. The methods section lacks some details about e.g. the polymerase used for the amplification step (some polymerases like AmpliTaq Gold have a bias towards shorter fragments) and purification method (beads vs. columns). And if beads were used, what bead:DNA ratio was used as this determines the size selection? In addition, the deamination rate is very sample specific and may depend on the specimen preparation and storage conditions in the herbarium. For example, Fig. 3b of Weiß et al. (2016) shows that some samples collected before 1900 can have C>T misincorporations below 2 % while others of the same age are above 4%. So to actually test differences between the two library methods, it would be better to use the same DNA extract of the same sample.

>> As explained to referee 1, the deamination rates, assessed in our study for 13 herbarium samples, were mainly a prior for ancient DNA authentication. After supplementary discussions with Fasteris, we now list in more detail the steps which are significantly different between each library protocol (Materials and Methods, lines 126 to 137). For each of the 13 samples, the totality of the material was either sent to Fasteris, or processed in our laboratory, for bead purification prior to library conversion. As this precludes the possibility of using a same sample for protocol comparison, we removed any "protocol effect" interpretation in results (line 309) and discussion (lines 462-467).

We believe that the size of the DNA fragments may be linked to deamination, for three reasons: i) The Truseq protocol includes two bead purification steps, in favor of shorter fragments than the ones used in the BEST protocol; ii) the size of the merged sequences obtained is significantly smaller for the "Truseq" protocol (when comparing fragment length categories, the BEST library was enriched in longer fragments (66-140 nt), while the TruSeq Nano library was enriched in smaller fragments (15-40 nt and 41-65 nt) (p-value<0.0001 in all cases, aov function, R package, data not shown); iii) extracting ultrashort DNA molecules from herbarium specimens, Gutaker et al.⁴ showed, comparing two different DNA binding methods on same samples, that a decrease in median fragment length was accompanied by an increase in C-to-T substitutions.

- L. 418: What polymerases were actually used for the amplification step? Did you use a polymerase incapable of recovering uracils?

>> We used U-readthrough polymerases (guaranteed but anonymous for libraries from Fasteris, and PFU Turbo Cx for BEST home-made libraries), as now stated in the Material & methods section lines 131-134.

Reviewer #4 (Remarks to the Author):

* General Comments

Campos et al. present a thorough investigation of historical and modern genomes reconstructed from the Citrus pathogen *Xanthomonas citri* pv. *citri*. Their results are very compelling, and I am impressed by the comprehensiveness of their analyses, from ancient DNA authentication and quality control all the way to Bayesian phylogenetic and phylogeographic modeling. I have only a small number of technical comments and minor concerns that I would like to see addressed. Overall however, I think the present study does a great job of highlighting the power and utility of herbarium genomics, especially to study plant pathogens.

** Methods

- This is more of a curiosity-driven question, but did you also map sequence reads to the suspected hosts? If so, was the host always correctly identified in the herbarium record? (I'm not proposing the authors to do this if they haven't, but it would be interesting to report if they did already do this analysis anyways.)

>> In this study, we decided to focus our analyses on the pathogenic bacterium *Xci*. In a previous work⁵, HERB_1937 reads were mapped to *Citrus* host reference genomes to assess the taxonomic composition of such an herbarium sample. Analyses aiming to use genomic data from all the available historical samples to determine host taxonomy and validate/correct the herbarium record are currently undergoing and will require further development before being published.

- Along similar lines, I think it's a little confusing to refer to the *Xci* DNA content as "endogenous" (e.g. in Table 2 and the Discussion), since really, the "endogenous DNA" of the specimen would be the host plant, with *Xci* being a pathogen infecting it

>> We removed any mention of "endogeneous DNA" in our revised manuscript.

- For merging aDNA paired-end reads, please also report the percentage of merged reads (in addition to median insert length after merging), since a low fraction of merged pairs can be indicative of long fragments present in the library which would be missed when reporting just the length statistics of the merged fraction

>> Percentage of merged reads were fairly high (between 96.97 and 99.97%) and are now reported in the Table 2 of our revised manuscript.

- Also, did you continue the data analysis with only the merged fraction of the herbarium libraries? This is unclear from the current Methods.

>> Yes, both the merged reads and the remaining non-merged ones were kept for subsequent analyses, as now specified lines 156-157.

- picardtools MarkDuplicates doesn't properly remove duplicates for merged aDNA reads. I'm not proposing to re-run everything, but it would be a helpful sanity check to compare the percentage of duplicates removed from picardtools vs. a tool optimized for aDNA such as DeDup: <https://github.com/apeltzer/DeDup>

>> We were not aware of the existence of such a tool. When including DeDup in our pipeline, we removed less PCR duplicates in historical reads than with MarkDuplicates (15 % vs 40%, in average), in accordance with the conceptual idea of this aDNA optimized method. As we filter SNPs based on depth, which would be influenced by the identification and removal of PCR replicates, we also compared high-quality SNPs identified using each tool. Out of the 15,281 SNPs identified on the whole dataset, 99.13% of them were shared (identified by both the two tools), comforting us in the minor impact of using MarkDuplicates instead of DeDup in our analysis (see below). We will definitely use DeDup in our future inferences.

- The authors use ClonalFrameML to remove recombinant regions from their dataset, as those violate the assumptions of their subsequent phylogenetic analysis. I would be interested in additional details as to how well this actually eliminates reticulations from their trees, for example by comparing phylogenetic networks before and after the ClonalFrameML-based removal (using SplitsTree, for example)

>> As suggested, we built phylogenetic networks with SplitsTree using two SNP alignments either excluding, or not, recombinant sites estimated with ClonalFrameML. The obtained results confirm that ClonalFrameML successfully identified and removed recombinant regions (as highlighted by differences both in the observed network reticulations and PhiTest statistical p-values).

SplitsTrees Networks

- Regarding the pathogenicity-associated gene content analysis: it appears to me that there may be quite a few homologous genes in the set of effectors you map to. It would be good to give a little more detail in the Methods section on how you handle the issues that may arise from this. Do you map to

these one-by-one? or all at once? how do you deal with situations where a read may map equally well to multiple genes? Do you compare the genic coverage to a measure of whole-genome coverage of the same individual to assess potential issues with copy-number or multiple mappings?...

>> Details were added in the methodology section. We intentionally chose to include in our set of effectors slightly divergent homologs and truncated or pseudogenized versions of some genes, notably because matching length is an important parameter in our analysis. In such cases each gene was examined individually and thoroughly by eye to address these issues. There was no case where reads mapped equally well to different homologues. Copy number was not assessed and is complicated by plasmid-carried effector genes, an example being TALE genes for which presence can be assessed but structure and number cannot be evaluated.

** Analyses

- Fig. 2A: there seems to be a mismatch between the slope reported in the figure inset vs. in the text (line 329).

>> This typo has now been corrected.

- It's a little confusing to have the BEAST analysis already mentioned when describing Fig. 2B, before the analysis is properly introduced for Figure 3. Maybe this could be restructured slightly to make more clear what analysis is done for Fig. 2B?

>> We removed the mention of any BEAST analysis in this section and now only refers to the "date-randomization test", previously described in the material & methods part.

- Table 3: it would be helpful to clarify that negative dates correspond to dates "BC" in this table (and elsewhere), to avoid confusion with dates "Before Present" that are sometimes presented in a similar fashion.

>> Dates in Table 3 are given in "Current Era" (CE) unit. This has been added both to table header and legend.

* Specific Comments

- I.72/73: Not sure it's clear to the reader what "quarantine organism status" is supposed to mean

>> We rephrased: "and by resulting in strong restriction of commercial exchange of fruits and plants from infested regions".

- I.92: It would be helpful for understanding to more clearly state what the "sampling covering 70 years" refers to here (it looks like it refers to the time frame during with the "modern" samples analysed here were sampled?)

>> We rephrased: "In the present study, we took advantage of an extensive collection of contemporary *Xci* strains sampled in the field during the last 70 years along with..."

- l.320: rephrase "Lineage A2 corresponds to lineage A2" -- confusing otherwise

>> We rephrased by adding a reference

- l.526: you mention 163 genomes here, but previously you refer to 171+13 genomes

>> 163 is the number of pathotype A *Xci* genomes. 171 + 13 is the total number of included *Xci* genomes (including A, A* & Aw pathotypes/lineages).

* Writing

- l.24: "shed light *on*..."

>> Done.

- l.196: "*using* three different tests..."

>> Done.

- l.376: "we *used/performed* a rate-dating analysis..."

>> Done.

- l.472: "*constrained*" instead of restrained

>> Done.

References cited:

1. Patané, J. S. L. *et al.* Origin and diversification of *Xanthomonas citri* subsp. *citri* pathotypes revealed by inclusive phylogenomic, dating, and biogeographic analyses. *BMC Genomics* (2019) doi:10.1186/s12864-019-6007-4.
2. Richard, D. *et al.* Time-calibrated genomic evolution of a monomorphic bacterium during its establishment as an endemic crop pathogen. *Mol. Ecol.* (2021) doi:10.1111/mec.15770.
3. Weiß, C. L. *et al.* Temporal patterns of damage and decay kinetics of dna retrieved from plant herbarium specimens. *R. Soc. Open Sci.* (2016) doi:10.1098/rsos.160239.
4. Gutaker, R. M., Reiter, E., Furtwängler, A., Schuenemann, V. J. & Burbano, H. A. Extraction of ultrashort DNA molecules from herbarium specimens. *Biotechniques* **62**, 76–79 (2017).
5. Campos, P. E. *et al.* First historical genome of a crop bacterial pathogen from herbarium specimen: Insights into citrus canker emergence. *PLoS Pathogens* (2021) doi:10.1371/journal.ppat.1009714.
6. Nielsen, R. & Signorovitch, J. Correcting for ascertainment biases when analyzing SNP data: applications to the estimation of linkage disequilibrium. *Theor. Popul. Biol.* **63**, 245–255 (2003).

7. Das, A. K. Citrus Canker—A Review. *J. Appl. Hortic.* **5**, 52–60 (2003).
8. Wu, G. A. *et al.* Genomics of the origin and evolution of Citrus. *Nature* **554**, 311–316 (2018).
9. Rieux, A. & Balloux, F. Inferences from tip-calibrated phylogenies: A review and a practical guide. *Molecular Ecology* vol. 25 1911–1924 (2016).
10. Robène, I. *et al.* Development and comparative validation of genomic-driven PCR-based assays to detect *Xanthomonas citri* pv. *citri* in citrus plants. *BMC Microbiol.* (2020) doi:10.1186/s12866-020-01972-8.

Reviewer #1 (Remarks to the Author):

The authors have addressed my concerns. Congratulations for the nice work.

Reviewer #2 (Remarks to the Author):

All my comments have been adequately addressed.

Reviewer #3 (Remarks to the Author):

The authors thoroughly revised their manuscript and responded well to my and the other reviewers comments. This manuscript is a great example of how the inclusion of historical herbarium specimens can be used for plant pathogen studies.

I greatly appreciate the extra work the authors did in response to the comments. I found only one small typo: I think "small reads" should be replaced with "short reads" (l. 435).

Reviewer #4 (Remarks to the Author):

All my comments have been addressed satisfactorily, and I'd like to again applaud the authors for their thoroughness and comprehensiveness.

Your manuscript entitled "Herbarium specimen sequencing allows precise dating of *Xanthomonas citri* pv. *citri* diversification history" has now been seen again by our referees, whose comments appear below. In light of their advice I am delighted to say that we are happy, in principle, to publish a suitably revised version in Nature Communications under the open access CC BY license (Creative Commons Attribution 4.0 International License). We therefore invite you to revise your paper one last time to address the remaining concerns of our reviewers and our editorial requests in the attached document(s). At the same time we ask that you edit your manuscript to comply with our policies and formatting requirements and to maximise the accessibility and therefore the impact of your work.

>> We are grateful to the reviewers and yourself for the final comments. Below, please find a point-by-point reply to all comments.

REVIEWER COMMENTS

Reviewer #1: The authors have addressed my concerns. Congratulations for the nice work.

>> Thanks

Reviewer #2: All my comments have been adequately addressed.

>> Thanks

Reviewer #3: The authors thoroughly revised their manuscript and responded well to my and the other reviewers comments. This manuscript is a great example of how the inclusion of historical herbarium specimens can be used for plant pathogen studies. I greatly appreciate the extra work the authors did in response to the comments. I found only one small typo: I think "small reads" should be replaced with "short reads" (l. 435).

>> Thanks, the typo has been corrected.

Reviewer #4: All my comments have been addressed satisfactorily, and I'd like to again applaud the authors for their thoroughness and comprehensiveness.

>> Thanks